# Sex chromosome gene expression associated with vocal learning following hormonal manipulation in female zebra finches

Matthew Davenport[1]*[†], Ha Na Choe[2†], Hiroaki Matsunami[2,3], Erich Jarvis[1,4]*

[1]Laboratory Language Neurogenetics, The Rockefeller University, New York City, United States; [2]Department of Molecular Genetics and Microbiology, Duke University School of Medicine, Durham, United States; [3]Department of Neurobiology, Duke Institute for Brain Sciences, Duke University School of Medicine, Durham, United States; [4]Howard Hughes Medical Institute, Chevy Chase, United States

*For correspondence:
mdavenport@rockefeller.edu (H);
ejarvis@rockefeller.edu (D)

[†]These authors contributed equally to this work

## eLife Assessment

This study is **valuable** as it provides information about the genes regulated by sex hormone treatment in song nuclei and other brain regions and suggests candidate genes that might induce sexual dimorphism in the zebra finch brain. The analysis presented is thorough and detailed. Whereas the evidence for gene regulation by hormone treatment is well supported, the evidence for an association of those genes with song learning (as written in the title) is **incompletely** supported as no manipulation of song learning or song analysis was conducted.

**Abstract** Zebra finches are sexually dimorphic vocal learners. Males learn to sing by imitating mature conspecifics, but females do not. Absence of song in females is associated with partial atrophy and apparent repression of several vocal learning brain regions during development. However, atrophy can be prevented, and vocal learning retained in females when given early pharmacological estrogen treatment. To screen for candidate drivers of this sexual dimorphism, we performed an unbiased transcriptomic analysis of song learning nuclei specializations relative to the surrounding regions from either sex, treated with vehicle or estrogen until 30 days of age when divergence between the sexes becomes anatomically apparent. Analyses of transcriptomes by RNA sequencing identified song nuclei-specialized gene expressed modules associated with sex and estrogen manipulation. Female HVC and Area X gene modules were specialized by estrogen supplementation, exhibiting a subset of the transcriptomic specializations observed in males. Female robust nucleus of the arcopallium (RA) and lateral magnocellular nucleus of the anterior nidopallium (LMAN) specialized modules were less dependent on estrogen. The estrogen-induced gene modules in females were enriched for anatomical development functions and strongly correlated to the expression of several Z sex chromosome genes. We present a hypothesis where reduced dosage and expression of these Z chromosome genes suppress the full development of the song system and thus song learning behavior, which is partially rescued by estrogen treatment.

## Introduction

Vocal learning is the ability to imitate heard sounds using a vocal organ and is a necessary and specialized component for spoken language and song. Vocal learning is found in seven nonhuman clades,

four mammalian and three avian, each having independently evolved the trait (*Jarvis, 2019*; *Jarvis et al., 2014*). Oscine songbirds have proved to be the most tractable for studying vocal learning in the lab, with much of the field focusing on the Australian zebra finch (*Taeniopygia castanotis*). Despite ~300 million years of separation from their common ancestor (*Kumar and Hedges, 1998*; *van Tuinen and Hadly, 2004*), there is remarkable evolutionary convergence between songbird and human vocal learning in terms of behavioral progression, developmental effects of deafening, anatomical connectivity of vocal-motor learning circuits, sites of accelerated evolution within the genome, and genes with specialized up- or downregulated expression in song and speech circuits relative to the surrounding motor control circuits (*Jarvis, 2019*; *Bolhuis et al., 2010*; *Doupe and Kuhl, 1999*; *Mooney, 2009*; *Pfenning et al., 2014*; *Feenders et al., 2008*; *Gedman et al., 2022*; *Cahill et al., 2021*; *Li et al., 2007*; *Lovell et al., 2008*). Unlike in humans, however, vocal learning is strongly sexually dimorphic in zebra finches and many other vocal learning species (*Nottebohm and Arnold, 1976*; *Odom et al., 2014*). Male zebra finches learn to produce a species-appropriate song by imitating mature male conspecifics during juvenile development, while females are limited to producing innate calls (*Zann, 1997*).

The songbird vocal motor learning circuit contains four major interconnected telencephalic song control nuclei: HVC (proper name) in the dorsal nidopallium (DN); the lateral magnocellular nucleus of the anterior nidopallium (LMAN) in the anterior nidopallium (AN); the robust nucleus of the arcopallium (RA) in the lateral intermediate arcopallium (LAI; also called AId); and Area X in the striatum (Str; *Figure 1a*; *Mooney, 2009*). During juvenile development in zebra finch females, HVC and RA atrophy, HVC fails to form synapses in RA, and Area X never appears (*Nottebohm and Arnold, 1976*; *Bottjer et al., 1985*; *Garcia Calero and Scharff, 2013*; *Konishi and Akutagawa, 1985*; *Nixdorf-Bergweiler, 1996*; *Nordeen and Nordeen, 1988*; *Shaughnessy et al., 2019*; *Holloway and Clayton, 2001*; *Mooney and Rao, 1994*). Amazingly, female zebra finches treated with estrogen or a synthetic analog at an early age do not exhibit song system atrophy and instead form a functional neural circuit with all the anatomical components and connections seen in males (*Gurney and Konishi, 1980*; *Gurney, 1982*; *Simpson and Vicario, 1991a*; *Simpson and Vicario, 1991b*). This 'masculinized' song system allows estrogen-supplemented females to imitate vocalizations, though not with the same accuracy as males (*Gurney and Konishi, 1980*; *Simpson and Vicario, 1991a*; *Choe et al., 2021*; *Pohl Apel, 1985*; *Pohl Apel and Sossinka, 1984*). Interestingly, lesioning female HVC prevents estrogen-dependent anatomical 'masculinization' of its postsynaptic targets to RA and Area X (*Herrmann and Arnold, 1991*).

The genetic basis of this estrogen-sensitive and sexually dimorphic vocal learning in zebra finch remains largely unknown beyond the downstream recruitment of the androgen receptor (AR) (*Nordeen et al., 1986*). However, the examination of a rare gynandromorphic zebra finch with lateralized sex chromosome composition indicates that genetically male HVC and Area X are larger than their female analogs independent of gonadal hormone production, implicating sex chromosome gene expression within the song system (*Agate et al., 2003*). Unlike the mammalian X and Y, birds have a Z and W sex determination system where females are hemizygous (ZW) and males homozygous (ZZ) (*Gianaroli et al., 2013*). The relevant transcriptional machinery appears to be set up by post-hatch day 30 (PHD30), after which estrogen fails to masculinize female song nuclei or behavior, and the male song system enlarges while the female song system atrophies (*Bottjer et al., 1985*; *Konishi and Akutagawa, 1985*; *Gurney and Konishi, 1980*; *Konishi and Akutagawa, 1988*). Taken together with the hypothesis that vocal learning in females was lost multiple independent times among songbirds (*Odom et al., 2014*), the extant findings suggest genetic drivers of vocal learning loss associated with estrogen, sex chromosomes, and song nuclei gene expression specializations. To screen for potential genetic drivers (loci whose expression/inheritance regulate the trait) and locate their action within the song system, we performed an unbiased analysis of transcriptomes from song nuclei and surrounding motor control regions in zebra finches of either sex chronically treated with 17-β-estradiol (E2) or vehicle from hatch until sacrifice at PHD30 (*Figure 1b*). While the birds for this study were sacrificed prior to the developmental presentation of song behavior, we have previously shown that female finches treated with E2 in the same exact way go on to produce rudimentary imitative songs as adults, consistent with the known induction of vocal learning in females by E2 (*Gurney and Konishi, 1980*; *Choe et al., 2021*). We used a new zebra finch genome assembly and annotation produced by the Vertebrate Genomes Project (VGP) containing both the Z and W chromosomes (*Rhie et al., 2021*).

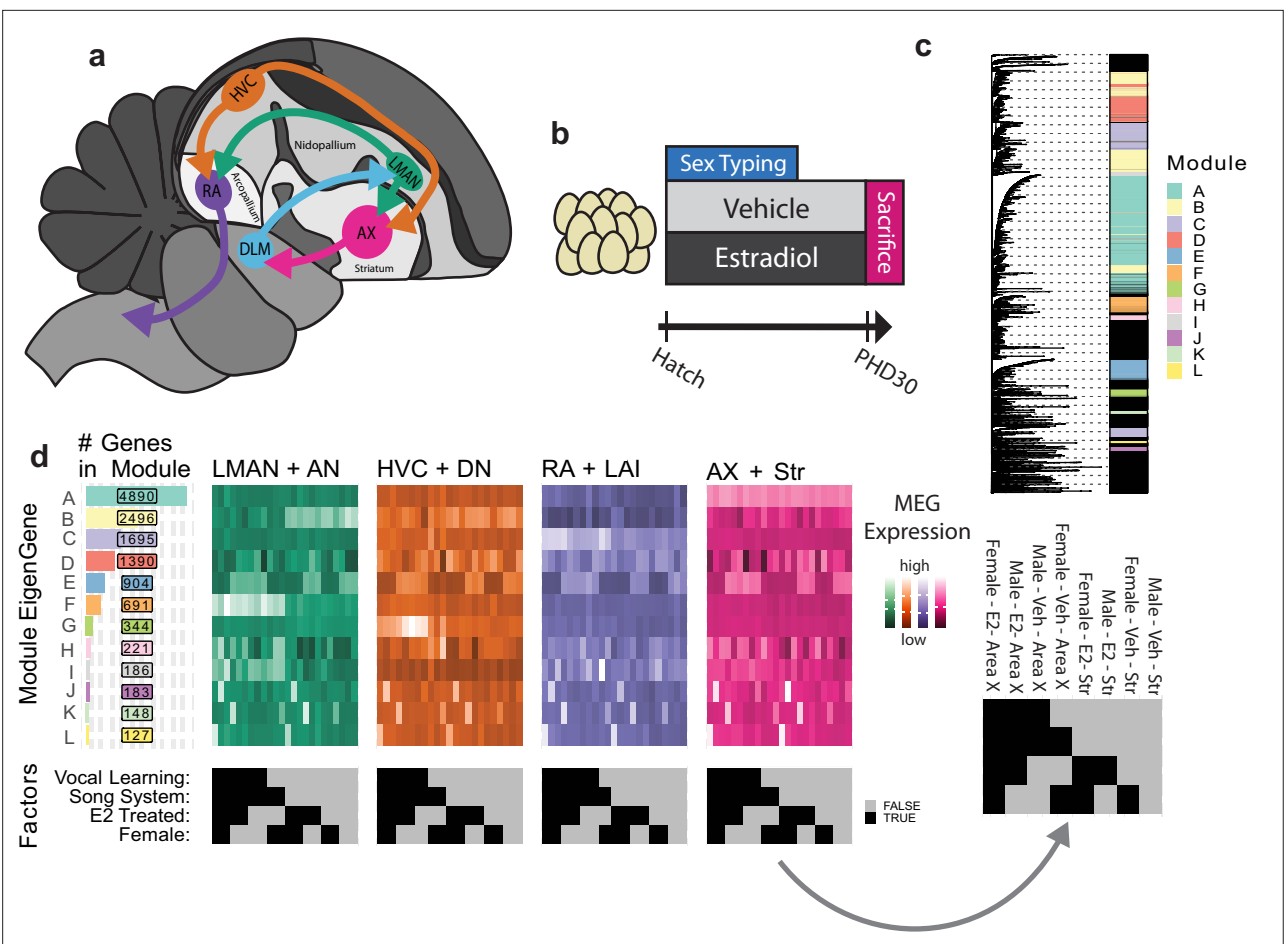

**Figure 1.** Song system anatomy and experimental design. (**a**) Diagram of song system connectivity within the adult male zebra finch brain with major telencephalic domains indicated. Area X connects back to the lateral magnocellular nucleus of the anterior nidopallium (LMAN) through the nonvocal-specific thalamic nucleus DLM. (**b**) Experimental design. Animals were treated with E2 or a vehicle from hatch until sacrifice on post-hatch day 30 (PHD30). (**c**) Weighted gene correlation network analysis (WGCNA) assignment of genes to modules. Left: Hierarchy computed over the transcriptome-wide topological overlap matrix of gene-to-gene correlations in transcript abundance across samples. Right: Module assignment raster, rows are genes colored according to the assigned module, unassigned genes in black. (**d**) Module eigengene (MEG) expression heatmaps arranged by module size (left) aligned to traits of interest (bottom). Each row is an MEG, and each sample is a column. Samples are grouped according to neural circuit node in different colored subpanels. Color intensity encodes MEG expression as calculated by WGCNA. An example raster with sample category labels is provided at right.

The online version of this article includes the following figure supplement(s) for figure 1:

**Figure supplement 1.** Outlier sample detection by hierarchical clustering.

**Figure supplement 2.** Selection of soft-thresholding power for weighted gene correlation network analysis (WGCNA) model.

**Figure supplement 3.** Selection of minimum module size and tree cut height parameter values for weighted gene correlation network analysis (WGCNA) model.

**Figure supplement 4.** Initial module overfitting to single samples.

## Results
### Identification of gene expression modules

We first sought to characterize differences in song nuclei gene expression specializations relative to their immediate surrounding motor brain regions in juvenile males and females with and without E2 treatment. We used RNA-seq data from a previous study by our lab, on the effects of E2 manipulation on the song system in both males and females (*Choe et al., 2021*). This E2 treatment program masculinized females sufficiently for them to produce song as adults in a parallel cohort of birds (*Choe et al., 2021*). Birds were sacrificed as juveniles at PHD30, after a 1 hr period of silence to

limit activity-dependent gene expression in song nuclei and surrounding motor regions; this developmental age was chosen because it is around the time when both males and E2-treated females start to sing (e.g. subsong), and all four song nuclei are sufficiently developed to be visually apparent in histological sections (*Choe et al., 2021*). The four major song nuclei (HVC, LMAN, RA, and Area X) and their adjacent surrounding motor regions (DN, AN, LAI, and Str, respectively; *Figure 1a*) were dissected using laser capture. In the case of vehicle-treated females, which lack Area X, a piece of striatum was taken from where Area X would be in males to serve as the Area X sample. In our previous study, we found that these PHD30 vehicle-treated males had larger RAs and HVCs than their female counterparts. Area X was absent in vehicle-treated females, and LMAN was similarly sized in both sexes. Following E2 treatment, E2-treated male RA was significantly smaller than vehicle-treated males, while Area X appeared in E2-treated females, and HVC was significantly larger than in vehicle-treated females (*Choe et al., 2021*). This past study mapped the RNA-seq reads to an older genome assembly lacking the W sex chromosome (*Choe et al., 2021*).

With this data, we first remapped RNA-seq reads (from n=3 birds per group) to a new zebra finch assembly (GCF_008822105.2) with both the Z and W chromosomes. As many W chromosome genes are duplicated from the Z chromosome and thus highly similar in sequence, only single-mapped reads were considered to minimize misattributed Z chromosome reads. In our quality control analyses, hierarchical clustering of sample expression vectors revealed two technical outliers, one HVC from a vehicle-treated male and one RA from an E2-treated female, which we excluded from further analysis (*Figure 1—figure supplement 1*). Similar to previous PCA and hierarchical clustering (*Choe et al., 2021*), the remapped RNA-seq data expression levels with outliers removed still resulted in separation of song nucleus and surround and some E2-treated female samples, without sufficient resolution at finer group levels (*Figure 1—figure supplement 1*). To address our questions at finer resolution and in an unbiased way, we performed weighted gene correlation network analysis (WGCNA), where we decomposed the transcriptome into actively expressed gene modules using hierarchical clustering of the gene adjacency matrix. This matrix describes the inferred structure of gene networks within our data and was calculated with a soft-thresholding of the matrix of gene-to-gene expression correlations across samples (*Figure 1—figure supplement 2*; *Langfelder and Horvath, 2008*). After the genes were given initial hierarchical cluster-based module assignment, they were then iteratively reassigned to the module whose aggregate expression they best correlated with until no additional genes met the WGCNA reassignment threshold (*Figure 1—figure supplement 3* and Materials and methods for parameterization). This reassignment was performed to ensure that genes are matched to the aggregate measure that best represents their expression.

Of the ~21,000 annotated zebra finch genes, 13,220 were well expressed in the finch telencephalic brain regions sampled, a comparable number of genes to what we have seen expressed in adult zebra finch telencephalon (*Gedman et al., 2021*; *Gedman, 2021*). These 13,220 were assigned to 14 co-expression modules (*Figure 1—figure supplement 4*). The median observation of unassigned transcripts was 22-fold lower than the median observation of module-assigned transcripts (4.86 vs 0.22 FPKM). Two modules clearly marked single samples from two different birds (*Figure 1—figure supplement 4a and b* - arrows), indicative of technical overfitting; these two modules (not birds) were excluded from further analysis, resulting in a more visibly diverse pattern within and across modules (*Figure 1—figure supplement 4c, d*). The remaining 12 modules contained 12,444 genes in total; these 12 modules were lettered in descending order of size A through L, containing from 4890 to 127 constituent genes each, which were dynamically expressed across brain regions and treatment groups (*Figure 1c*). The results of module assignment can be found in *Supplementary file 1*.

## Song nuclei specialization modules and sex differences

To understand song system transcriptional specialization in the context of the gene modules, we calculated the module eigengene (MEG) expression values for each of the 12 modules (*Figure 1d*). MEGs are the first principal component of variance of all genes in a module and are the aggregate measure for each module's expression across samples. We then tested for statistically significant correlations between MEG expression for each module and the song nuclei specializations relative to their respective surrounds. Each song nucleus in vehicle-treated control males had unique and overlapping specialization of genes in 2–4 modules of the 12 (*Figure 2a*). The male LMAN specialization relative to the surrounding AN was correlated with modules B, F, and I; male HVC specialization

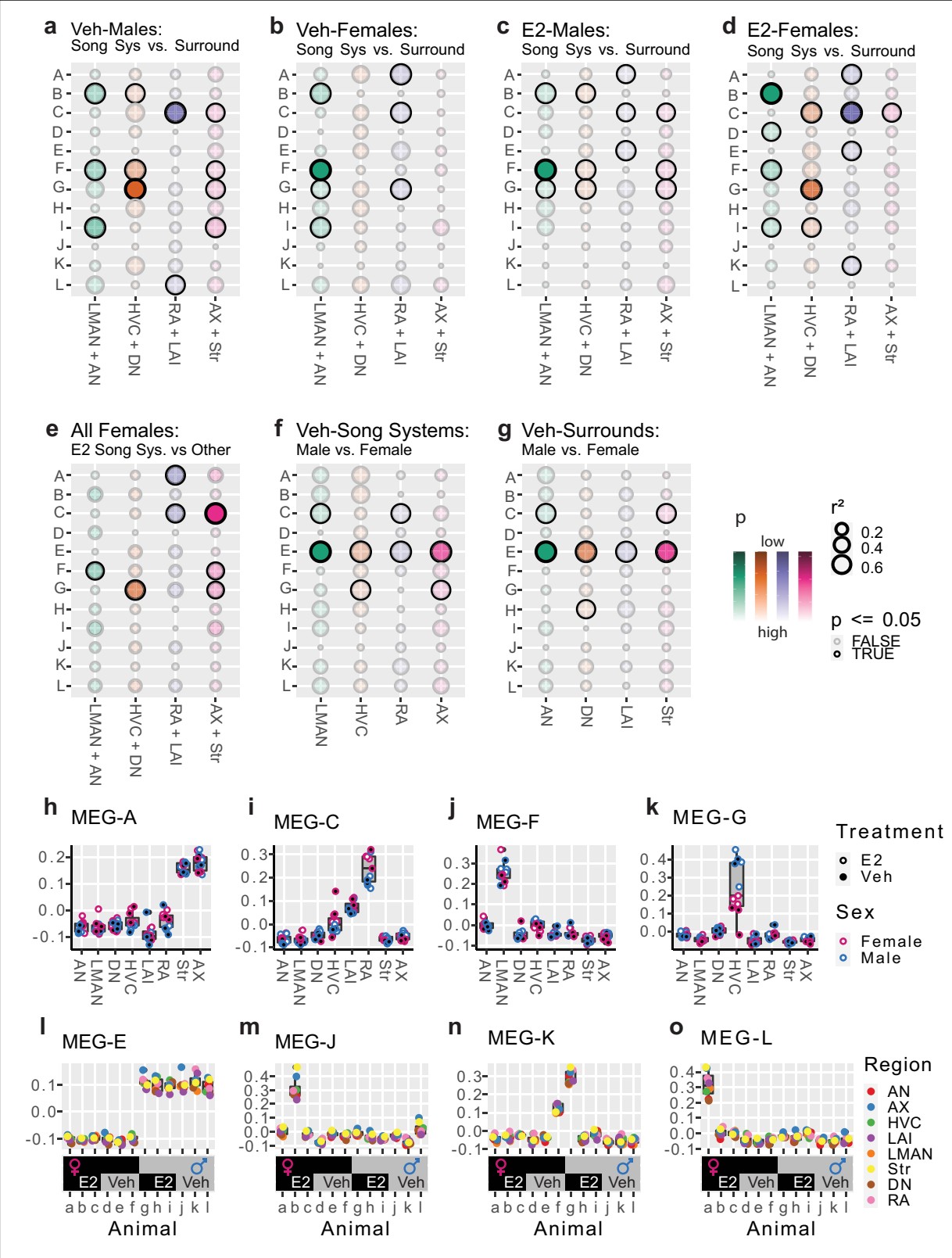

**Figure 2.** Association of modules to experimental variables. (**a–g**) Bubble plots showing statistical association between module eigengene (MEG) expression and variables of interest in various sample subsets. Strength of association (r²) is encoded by bubble size, significance (p) is encoded in the color scale with significant associations darkly bordered. Pearson's correlation and Student's t-test, alpha = 0.05. Plots show the associations between gene modules (rows) to: **a–b** vehicle-treated song system specializations, comparing MEG expression in song system samples from either

*Figure 2 continued on next page*

*Figure 2 continued*

sex to their appropriate surrounding controls; **c–d** E2-treated song system specialization, same comparison as **a–b** but within E2-treated samples; **e** female vocal learning capacity after E2, comparing E2-treated female song system components to all other female samples from that circuit node; **f** sexual dimorphism within the song system, comparing vehicle-treated male and female song system components; **g** sexual dimorphism within the surrounding control regions, comparing the vehicle-treated male and female surrounding control samples. Each neural circuit node is considered separately (columns). (**h–k**) Expression of modules with strong region-specific expression. Module A is highly expressed in Str and Area X samples, with additional differences between robust nucleus of the arcopallium (RA) and lateral intermediate arcopallium (LAI) (**h**). Module C is highly expressed in the arcopallium, especially RA, with some increase in HVC (**i**). Module F is expressed highly only in the lateral magnocellular nucleus of the anterior nidopallium (LMAN) regardless of sex or treatment (**j**). Module G is only highly expressed in HVC, where it differs by both sex and treatment (**k**). (**l–o**) Expression of selected MEGs by animal (top) aligned to their respective experimental variables (bottom), color indicates region. The sex chromosome enriched module E was highly expressed in all male samples and depleted in all female samples regardless of brain region or pharmacological treatment. Module J, K, and L eigengenes were each highly expressed in samples from one (J and L) or two (K) animals across all brain regions sampled.

relative to DN consisted of genes in modules B, F, and G; male RA specialization relative to LAI consisted of genes in modules C and L; and male Area X specialization relative to the surrounding Str consisted of genes in modules C, F, G, and I. Area X was the only song nucleus that did not have a specialized module unique to it relative to other song nuclei (*Figure 2a*). In nonvocal learning juvenile females, interestingly, LMAN was specialized relative to AN by the same gene modules as in males (B, F, and I) as well as an additional module G (*Figure 2b*); RA was specialized by module A as in males, but not module L and by additional modules A and G. In contrast, neither juvenile female HVC nor Area X exhibited significant gene module expression specializations relative to their surrounds.

## E2-responsive gene modules in song nuclei

We next assessed the effects of chronic exogenous estrogen on the developing song system. In the E2-treated juvenile males, the song nuclei-specialized modules overlapped with those seen in vehicle-treated males (*Figure 2a vs c*). Differences were that: in LMAN and Area X, module I was no longer present; in RA, module L was no longer present, module A appeared as seen in females, and module E also appeared. These differences in gene module specializations in the male song system are consistent with the E2 treatment regimen used, causing a slight decrease in vocal learning accuracy in males (*Choe et al., 2021*).

In contrast, E2-treated females more closely mirrored the gene module specializations seen in the vehicle-treated male song system (*Figure 2a vs d*). Specifically, in LMAN, modules B, F, and I were retained similar to vehicle-treated males, and module D uniquely appeared; in HVC, module G expression appeared and was strongly specialized as in vehicle-treated males, but modules C and I appeared relative to vehicle-treated females (*Figure 2b vs d*); in RA, module G disappeared and module C was retained as in vehicle-treated males, and module A was retained and module K appeared relative to vehicle-treated females; and in Area X, only module C appeared specialized in the E2-treated females. That is, the transcriptional response of female song nuclei to E2 treatment appeared to be far more dramatic than in males.

We performed an additional test for E2-induced changes to gene module expression in females by comparing E2-treated song nuclei in females to the combination of E2-treated surrounds from the same animals, vehicle-treated female surrounds, and vehicle-treated female song nuclei (*Figure 2e*). This analysis represents a comparison between samples from vocal learning-capable female samples and nonvocal learning female samples at each node. It revealed 1–3 modules in each song nucleus of E2-treated females that were a subset of some of the strongest correlated modules found in control males (*Figure 2a vs e*). These were: module F in LMAN; module G in HVC; module C in RA; and modules C, F, and G in Area X. That is, these are four core modules for song nuclei that are the most sensitive to E2 induction or enhancement in females and most associated with the presence of vocal learning in females.

## Gene modules for telencephalic sex differences

We next tested whether some modules could be explained by sex differences in the brain, regardless of song nuclei presence or vocal learning status. We compared vehicle-treated male and female song nuclei and surrounds separately and found that module E strongly correlated with higher expression in all male brain regions relative to females (*Figure 2f, g*). Among the song nuclei, module G eigengene

expression was significantly higher in male HVC and Area X; conversely, module C expression was higher in female LMAN and RA (*Figure 2f*). Among the surrounds, module H eigengene expression (not significant in any other comparison) was significantly higher in female DN, and module C was significantly higher in female AN and Str (*Figure 2g*). These findings indicate that module E contains genes that could be explained by a broad transcriptome sex difference in the brain, whereas the expression of other modules is more specific to brain region and/or treatment.

## Gene modules with region- and bird-specific expression

We next quantified the magnitude of module expression differences and observed several gene modules with clear region-specific expression patterns (*Figure 2h–k*). Module A, which appeared specialized in RA relative to LAI in several comparisons, was highly expressed in all Area X and Str samples, regardless of treatment or sex (*Figure 2h*). Interestingly, *FOXP2*, a gene critical for spoken-language and vocal learning in humans (*Lai et al., 2001*) and songbirds (*Teramitsu and White, 2006*), was a potent member of module A, correlating with the eigengene at $r^2=0.92$ across all samples. Module C, which was specialized to RA, HVC, and Area X relative to their surrounds and sexually dimorphic in LMAN and Str, was most highly expressed in arcopallial samples, especially RA, with some increase in HVC (*Figure 2i*). Module F, which was specialized to LMAN relative to its surround in all such comparisons, was highly expressed exclusively in LMAN regardless of sex and E2 treatment (*Figure 2j*). Module G, which was specialized to HVC in a sex- and E2-dependent manner, was only highly expressed in HVC samples, with males having higher expression than females, and females further split by treatment (*Figure 2k*).

We also checked if genes in specific modules were enriched in their expression in specific animals or divisions of animals regardless of region. The module E eigengene was highly expressed in all male samples and lowly expressed in all female samples regardless of brain region or treatment (*Figure 2l*), consistent with brain sex differences (*Figure 2f, g*). At the other extreme, three small modules showed higher expression specific to individual birds: module J was highly expressed in all samples of animal 'b', an E2-treated female (*Figure 2m*); module K was highly expressed across all samples of animal 'f' vehicle-treated female, and at roughly half-dose in animal 'g', an E2-treated male (*Figure 2n*); module L was highly expressed in animal 'a', another E2-treated female (*Figure 2o*). These findings indicate that the three smallest modules, J, K, and L, although with some patterning to RA for the later two (*Figure 2a, d*), are strongly animal-specific in their expression. As we can think of no source of technical variation that would produce a broadly distributed shift in the neural expression of a single gene module, this variation is likely attributable to biological interindividual variation. Whether this is genetic or from life history, we cannot say from the present data.

## Functional enrichment of specialized modules

We sought to understand the cumulative biological function of genes among the modules. To do this, we mapped the zebra finch genes to their 1:1 human orthologs where possible and then used human gene annotation to examine the Gene Ontology (GO) functions enriched within each module's constituent genes. Of the 12,444 module-assigned genes, 7909 (63%) had 1:1 human orthologs annotated in Ensembl. We found GO terms significantly enriched in module G, which included 'DNA binding transcription factor activity', 'cell differentiation', 'anatomical morphogenesis', 'cell-to-cell signaling', and 'positive regulation of multicellular organism growth', indicating that module G genes specialized in male and E2-treated female HVC and Area X potentially act to integrate and differentiate late-born neurons. Other significantly enriched terms were 'extracellular matrix structural component', 'external side of the plasma membrane', and 'extracellular space', indicating that this module may also act to restructure the extracellular matrix, perhaps to accommodate new cells. Module E, which was differential between the sexes for both song nuclei and surrounds, had six significantly enriched terms, of which three pertained to DNA damage repair: 'nucleolus', 'transcription, DNA templated', and 'U2-type precatalytic spliceosome'. These results in module E indicate that there is a sexually dimorphic gene expression program broadly distributed across the finch telencephalon that likely acts within the nuclear environment. The full table of GO enrichments by module can be found in supplemental data (*Supplementary file 2*).

## Gene modules enriched for human speech-associated genes

We next asked if any of the modules were enriched for genes previously determined to be convergently specialized in songbird song nuclei and human speech brain regions (*Pfenning et al., 2014*; *Gedman et al., 2022*). These gene lists included the transcriptional convergence between RA and human dorsal laryngeal motor cortex (dLMC), HVC and dLMC, LMAN and dLMC, Area X and the anterior caudate, and Area X and the anterior putamen (*Gedman et al., 2022*; *Figure 3B, D*; *Feenders et al., 2008*). We found that module B, specialized to both LMAN and HVC (*Figure 2a*), was enriched for the convergently specialized genes in human dLMC (*Figure 3a*) known to match the upper layers of the cortex (*Feenders et al., 2008*). Module C, specialized to RA (*Figure 2a , i*), was enriched for the genes convergently specialized in dLMC and RA (*Figure 3a*) known to match the lower layers of the cortex (*Feenders et al., 2008*). Module A, highly expressed in Area X and Str (*Figure 2h*), was enriched for genes convergently specialized in Area X and human anterior striatum (both caudate and putamen; *Figure 3a*). Module I, specialized to Area X (*Figure 2a*), was also even more strongly enriched for the same convergences as module A (*Figure 3a*). These findings indicate that the genes previously identified as convergently regulated between songbird song brain regions and human speech brain regions are components in the larger specialized gene networks identified here using WGCNA.

## Gene modules enriched for specific chromosomes

We next determined whether any modules were enriched in genes from specific chromosomes. We performed a bootstrapped enrichment analysis, randomizing the mapping between genes and modules 50,000 times to approximate null distributions. p-Values for each chromosome-module pairing were then FDR-corrected. To do this as accurately as possible, we performed this analysis using revised chromosomal assignments and structure from the most recent zebra finch genome assembly, bTaeGut_1.4.pri, which better resolves the microchromosomes (*Kim et al., 2021*). Surprisingly, we found that each of the 12 modules was enriched for genes from at least one chromosome (*Figure 3b*). The most striking was module E genes, which were enriched on the Z and W sex chromosomes, with nearly all W expressed genes and ~⅔ of Z expressed genes being members of module E. This is consistent with module E exhibiting higher expression in male samples regardless of treatment or region (*Figure 2d, e*). For the autosomes, we observed significant enrichments of two HVC specialized gene modules on chromosome 1A: module B (specialized in male HVC and human dLMC) and module G (specialized in male HVC and E2-treated female HVC and sexually dimorphic in vehicle-treated HVC). These results are particularly interesting given that zebra finch chromosomes 1 and 1A are believed to be the result of a songbird-specific fragmentation of the ancestral chromosome 1 found in chickens (*Itoh and Arnold, 2005*; *Warren et al., 2010*).

Module A, the largest module, which was highly expressed in Area X and adjacent striatum, was enriched across five macrochromosomes (chr1, 2, 3, 4, and 10). Module F, which was strongly expressed in LMAN, was enriched on chromosomes 2 and 7. Modules C (most strongly expressed in the arcopallium and part of RA and Area X specializations) and I (a component of the LMAN specialization) were enriched on chromosome 6 and microchromosome 35. Module D, a component of the LMAN specialization in E2-treated females, was enriched across eight small microchromosomes (chr22, 23, 25, 27, 28, 29, 30, 31). Most (4 of the 5) of the smallest modules in gene counts were enriched in the microchromosomes: Module H, which was sexually dimorphic only in DN, was enriched on microchromosome 28; module K was significantly enriched on microchromosome 37 and in the 'other' category, which includes all remaining unnamed DNA scaffolds in the assembly, such as further microchromosomes; module J was also enriched in 'other'.

## Sex chromosome gene expression across regions

To better understand the relationship between the sex chromosomes and module E, we examined the distribution of membership in module E with the sex chromosomes separated. WGCNA allows us to consider gene membership in a module as a continuous variable, rather than a binary variable, by correlating each gene's expression profile to the MEG. Doing this for module E, we found that Z transcripts were positively correlated to the MEG while W transcripts were anticorrelated (*Figure 4e*). This is consistent with Z chromosome transcripts being generally lower expressed in females relative to males, while W chromosome transcripts were only expressed in female brains. This general reduction

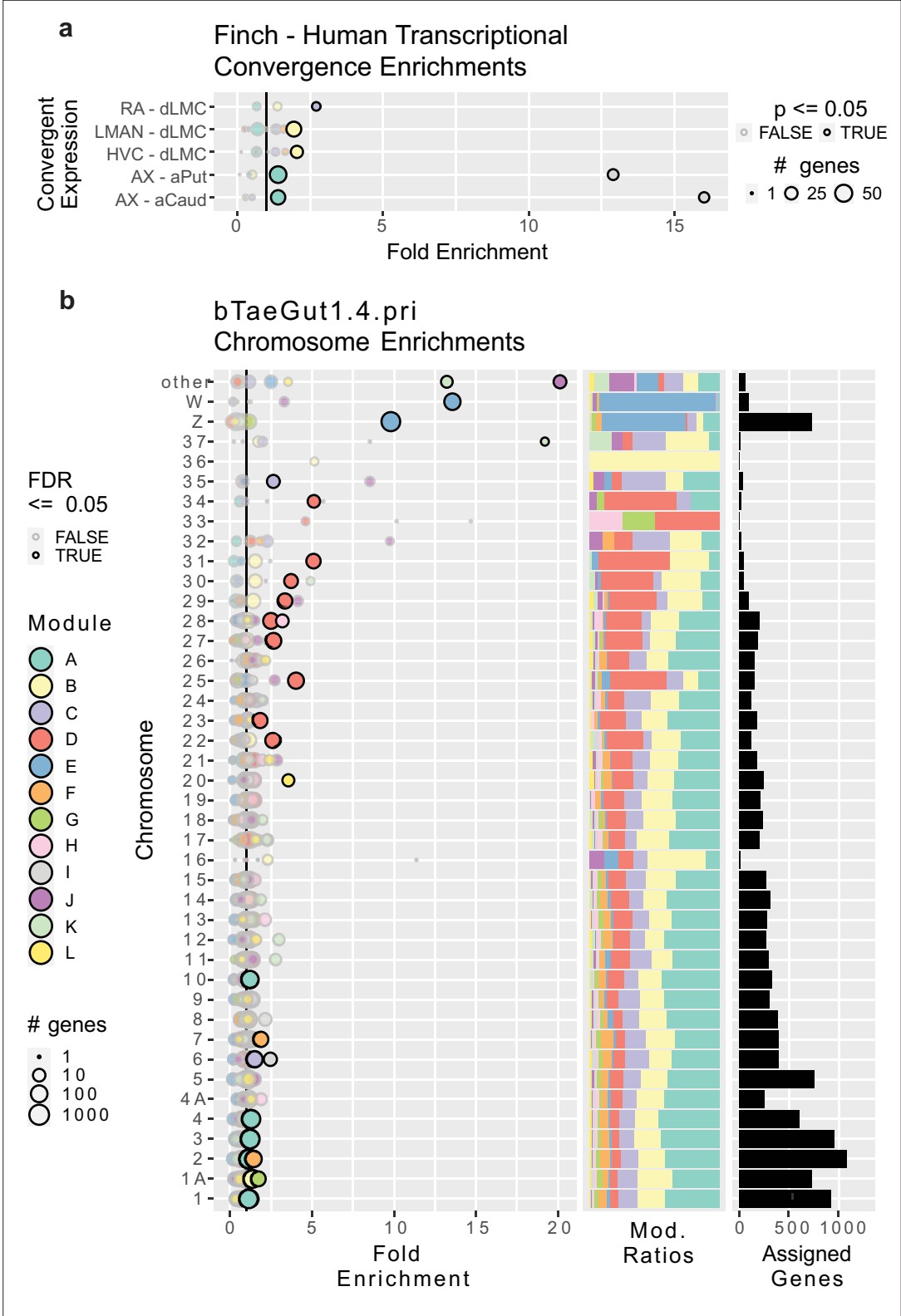

**Figure 3.** Gene module enrichments for human convergent signature and for chromosomes. (**a**) Enrichment of genes previously found to be convergently differentially expressed in the human laryngeal motor cortex and the pallial song nuclei or convergently expressed between the human vocal striatum and Area X. Bubble size linearly encodes the number of genes in each convergence module pairing. Significance was assessed using a one-tailed generally applicable gene set enrichment (GAGE) test, similar to Gene Ontology (GO) ontologies, alpha = 0.05.

*Figure 3 continued on next page*

*Figure 3 continued*

Significant enrichments are darkly bordered and opaque. Values to the right of the vertical black line indicate above random chance. (**b**) Enrichment of genes from specific chromosomes. Left, fold enrichment of modules onto zebra finch chromosomes in the newest genome assembly available; center, the portion of module-assigned transcripts from each chromosome per module; right, the number of module-assigned genes per chromosome. Each row is a chromosome, with each bubble representing the enrichment of transcripts from that chromosome in one of the gene modules defined by weighted gene correlation network analysis (WGCNA). Values to the right of the vertical black line indicate above random chance. The size of the bubbles indicates the log10 transformed number of genes in each chromosome module pairing. Significance was assessed using an FDR-corrected bootstrapped test of observed enrichment for each module chromosome pairing based on 50,000 randomizations of genes into modules. Significant enrichments are darkly bordered and opaque. a–b use the same color scale for modules.

in female Z chromosome transcript abundance within module E is consistent with the finding that diploid Z chromosomes in male birds do not have one copy inactivated to compensate for gene dosage, unlike the X inactivation in female mammals (*Agate et al., 2003*; *Itoh et al., 2007*).

Given the sexually dimorphic expression of module E across brain regions and the enrichment of the sex chromosomes within that module, we separately compared the expression of sex chromosome

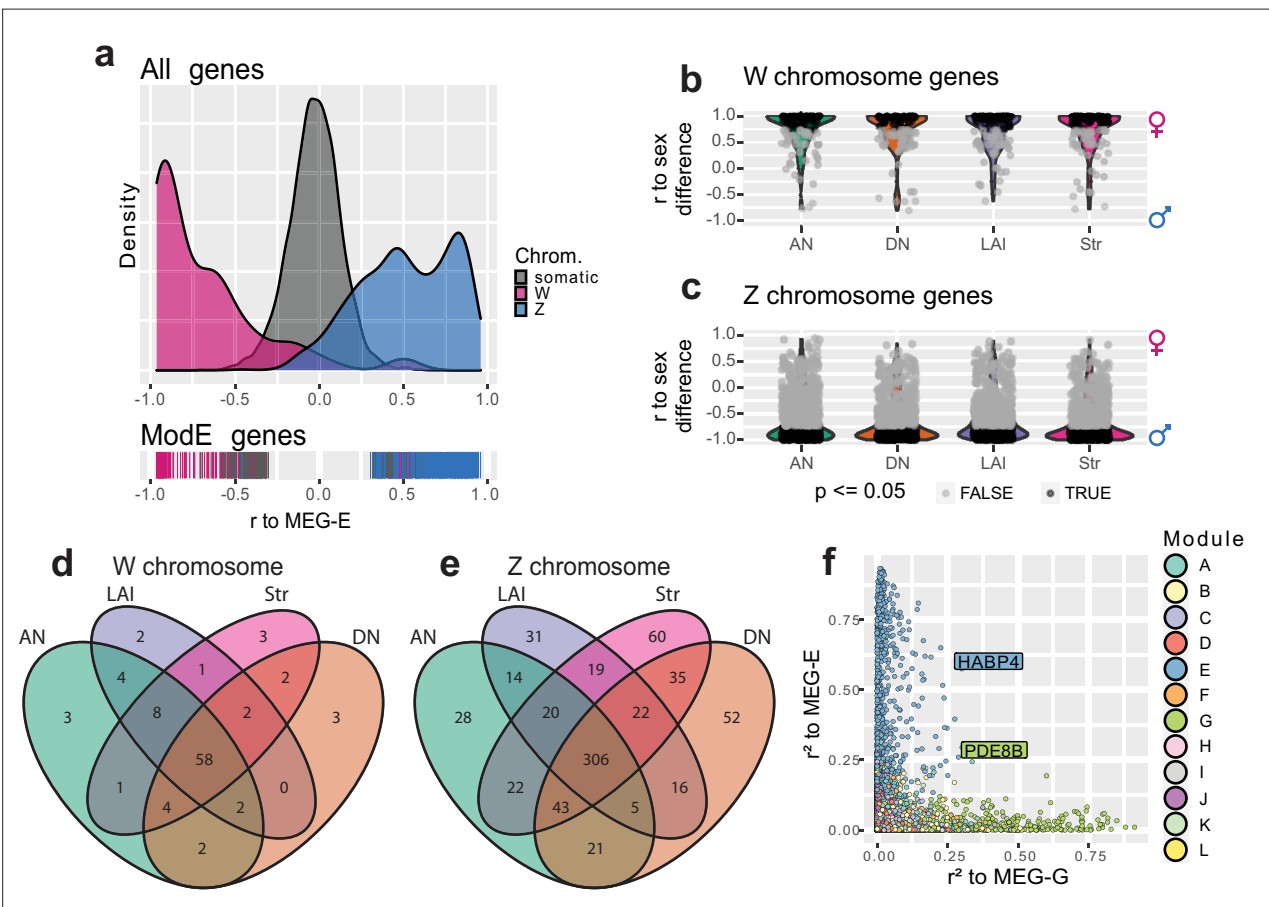

**Figure 4.** Brain-wide signatures of sex chromosome expression. (**a**) Distribution of continuous membership in module E across all module-assigned genes (top) and module E-assigned genes (bottom) based on correlation of expression to the module eigengene (Pearson's r to MEG-E) with sex chromosomes separated. (**b–c**) Distribution of sex chromosome gene expression correlations to the sex difference in vehicle-treated finches. Positive correlations indicate female-biased expression, while anticorrelations indicate male-biased expression. Significance was assessed in each region using an upper-tailed Student's correlation test for W chromosome transcripts (**b**) and lower-tailed for Z chromosome transcripts (**c**) with significant correlations in black, alpha = 0.05. (**d–e**) Venn diagrams intersecting the significantly sex difference correlated genes across nonvocal surround regions for the W and Z chromosomes respectively. (**f**) Comparison of continuous membership in module E (r2 to MEG-E, y-axis) and module G (r2 to MEG-G, x-axis) across all 12,444 module-assigned genes.

genes in the nonvocal motor surrounds of vehicle-treated control males and females regardless of WGCNA assignment to better understand sex chromosome expression without the influence of vocal learning specialization or E2 response. To do this, we first tested for correlated expression between each sex chromosome transcript to the animals' sex within each region such that female-enriched transcripts were positively correlated and male-enriched transcripts were anticorrelated with sex (*Figure 4f, g*). In these gene sets, we identified between 73 and 82 significantly expressed W chromosome genes and between 433 and 527 significantly depleted Z chromosome genes for each of the surround regions in males relative to females. Examining the union of these gene sets revealed that a total of 95 W and 694 Z chromosome genes were differentially expressed between sexes in at least one brain region, 62% and 65% of annotated sex chromosome genes, respectively. Conversely, the intersection of these regional gene sets contained 58 significantly expressed W chromosome genes and 306 significantly depleted Z chromosome genes in all nonvocal regions in females, representing 38% and 29% of annotated sex chromosome genes, respectively (*Figure 4h, i*, *Supplementary file 3*). This indicates that there is both a large broadly distributed set of sex-enriched/depleted sex chromosome genes and regional patterned sex chromosome gene expression.

## Modeling vocal learning and sex chromosome module interactions

As it is unlikely that the gene modules act independently of each other, we sought putative interacting genes between two modules of interest, module G specialized in HVC and Area X and module E dominated by sex chromosome genes. To do this, we again replaced binary in-or-out module membership with continuous module membership by correlating each gene' expression to the relevant MEG (*Langfelder and Horvath, 2008*). This allowed us to quantify the extent to which any gene was associated with any module, regardless of initial assignment (*Nordeen et al., 1986*). Looking across all assigned genes for our modules of interest, we identified two outlier genes, *PDE8B* and *HABP4*, which were the most E-associated gene assigned to module G and the most G-associated gene assigned to module E, respectively (*Figure 4j*). Both genes were significantly upregulated in male HVC relative to its surround, but not in female HVC of either treatment. *PDE8B* catalyzes the hydrolysis of the second messenger cAMP, and mutations to the gene cause an autosomal dominant form of striatal degeneration in humans (*Appenzeller et al., 2010*). *HABP4* is an RNA binding protein, known to repress the expression and subsequent DNA binding of *MEF2C* (*Kobarg et al., 2005*), a Z chromosome transcription factor which has undergone accelerated evolution in songbirds (*Cahill et al., 2021*) and whose repression by *FOXP2* is critical for cortico-striatal circuit formation in mice related to vocal behaviors (*Chen et al., 2016*). Both *PDE8B* and *HABP4* are found on the Z chromosome, and *HABP4* was one of the 694 Z transcripts significantly reduced across all brain regions in vehicle-treated females relative to males (*Supplementary file 3*). These findings again indicate that Z chromosome genes may be subject to multiple gene regulatory programs: the broadly distributed brain transcript reduction driven by reduced sex chromosome copy number and the specialized upregulation in HVC for vocal learning behavior.

## Sex chromosome dosage effects by module

To better understand the relationship between gene modules, sex chromosome gene expression levels, sex chromosome dosage, and song nuclei specializations, we directly compared the abundance of Z and W transcripts between vehicle-treated male and female samples in surrounds and song nuclei. After averaging across samples for each brain region and combining them for all brain regions, Z chromosome transcripts were present in both females (pink) and males (blue) as expected (*Figure 5a*). Although more Z chromosome genes were assigned to module E, many were assigned to the other modules (*Figure 5a*), consistent with our chromosome mapping without enrichment analysis (*Figure 3b*). However, there were clear male vs female expression differences apparent for Z chromosome genes in module E, but no obvious sex differences for Z chromosome gene expression in other modules. In contrast, W chromosome transcripts were mainly expressed in females (pink) and not males (blue) and were generally restricted to module E (*Figure 5b*), also consistent with our chromosome enrichment analysis (*Figure 3b*). To further quantify these effects outside and inside of the song system, we computed the percent of total expression for each sex chromosome gene which came from male samples (male_avg/(male_avg +female_avg)) and compared these distributions to those predicted by chromosome dosage. The expected average percentages based upon dosage are

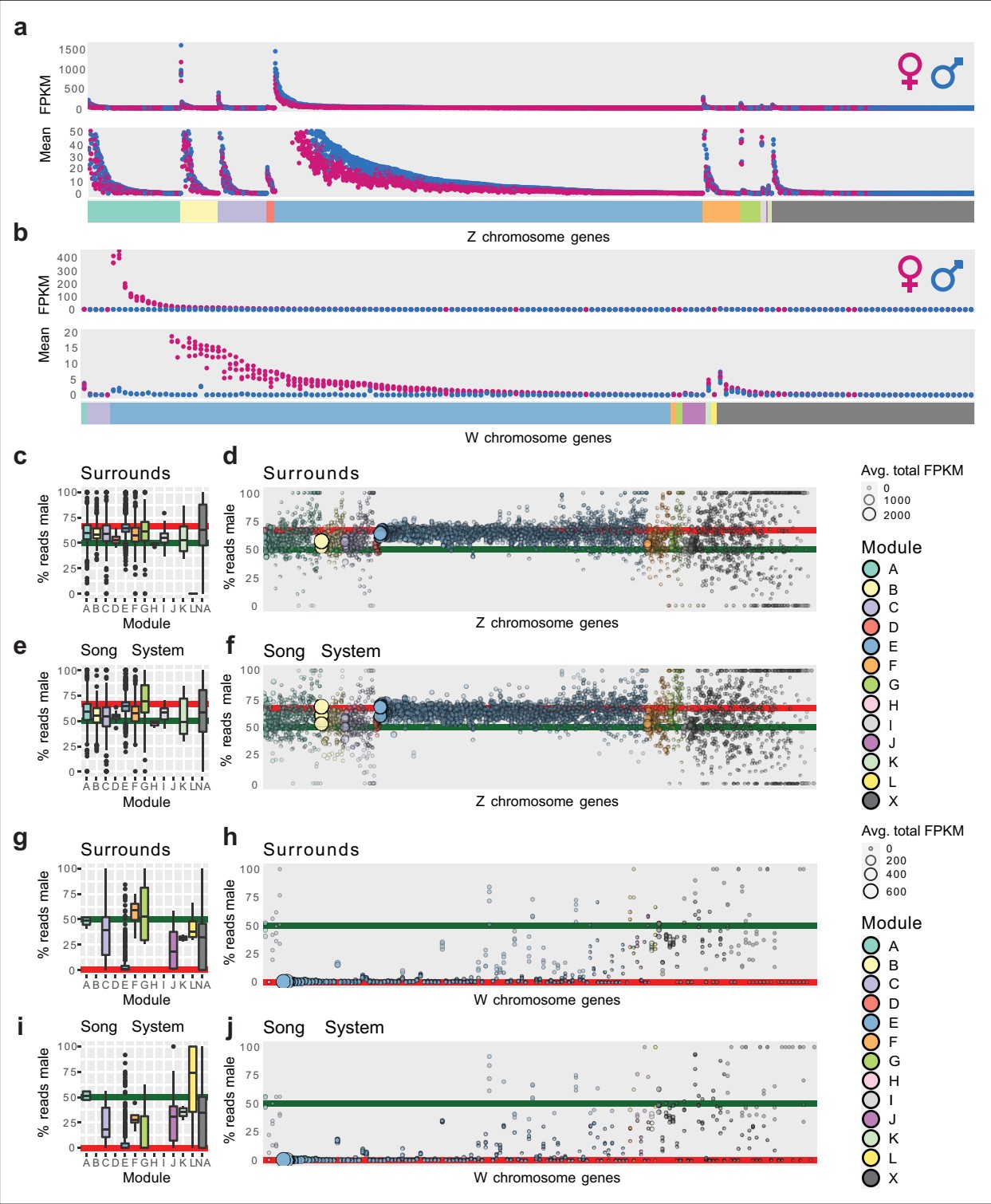

**Figure 5.** Sex chromosome gene module effects by module. (**a–b**) Scatter plots of Z and W chromosome transcript abundance averages for each region. X-axis contains all annotated sex chromosome genes ordered by module assignment (raster) and then expression (y-axis). Each gene has eight measured points, each the average of one brain region in vehicle-treated birds broken out by sex (color). Two y-axis scales are presented for each chromosome to help show the lowly expressed genes. (**c**) Boxplots describing the distribution of the percent reads from male samples per Z chromosome gene from **a**. (**d**) Bubble plot of the underlying Z chromosome single gene data from **b**; X-axis is ordered as in **a** with module assignment encoded by bubble color. The cumulative average expression is indicated by bubble size and opacity (male avg. FPKM + female avg. FPKM) with higher expressed genes being larger and more opaque. Red lines indicate the male read percentage expected for Z chromosome genes, 66.6%. Green

*Figure 5 continued on next page*

*Figure 5 continued*

lines indicate equal expression between sexes. Note that the Z chromosome genes of module E are expressed on average at almost exactly the sex ratio predicted by dosage, while Z chromosome genes in other modules show some degree of compensation. (**d–e**) Same as **b–d** but for song nuclei. (**f**) Boxplots describing the distribution of the percent reads from male samples per W chromosome gene from **b**. (**g**) Bubble plot of the underlying W chromosome single gene data from **f**; X-axis is ordered as in **b** with module assignment encoded by bubble color. The cumulative average expression is indicated by bubble size and opacity (male avg. FPKM + female avg. FPKM) with higher expressed genes being larger and more opaque. Red lines indicate the male read percentage expected for W chromosome genes, 0%. Green lines indicate equal expression between sexes. (**h–i**) same as b–d but for song nuclei.

66.6% male expression for Z chromosome genes (two Zs in males vs one Z in females; *Figure 5c–f*, red lines) and 0% male expression for W chromosome genes (0 W in males vs 1 W in females; *Figure 5g–j*, red lines).

Module E-assigned Z chromosome genes were expressed at almost exactly the ratio predicted by Z chromosome dosage on average; 65.1% measured vs 66.6% predicted, and the most abundantly expressed transcripts (largest circles) were expressed closest to the dose prediction in both surrounds (*Figure 5c, d*) and song nuclei (*Figure 5e, f*). For non-E modules, their Z chromosome transcripts were intermediate between equal expression and the prediction from chromosomal dosage in surrounds (*Figure 5c, d*), with Z chromosome genes in modules D and H being the least male-biased across regions (*Figure 5e, f*). One major difference between surrounds and song nuclei was the Z chromosome gene expression levels again in module G. The 25 Z chromosome genes in module G were more male biased in song nuclei compared to surrounds (*Figure 5c,d vs e,f*) and were the only set of Z chromosome genes that were male biased above the predictions from dose on average (*Figure 5e*). These findings indicate that the expression levels of Z chromosome genes in module E were predominantly dose regulated, with the abundance of the RNA dictated by the presence vs absence of those specific chromosomes regardless of brain region sampled. However, Z chromosome genes in the other modules whose expression is enriched for one or several brain regions, animals, or treatments exhibited varying degrees of dosage compensation to overcome the Z chromosome difference.

In contrast, W chromosome genes assigned to module E had on average 6.8% male read mapping counts (median 0.5%; *Figure 5g, h*). Non-module E-assigned W genes had 33.0% male reads mapped (median 32.7%; *Figure 5g, h*). As the values should be 0% reads from males mapped to the W chromosome, we believe this male read mapping is from less divergent transcript paralogs from the Z chromosome that map to the W chromosome. Examining the data at the level of single genes, we observed that module E-assigned W genes were far more likely than non-module E genes to show no or little putative Z paralog mapping (more genes on or near the 0% line; *Figure 5g, h*). Within module E, the genes with the lowest expression (smaller circle size) were the most likely to have putative paralogous expression (the lowest ~⅓ of these genes contributed to most of the expression above 0%, *Figure 5h*). We observed similar results in module E vs other modules for W expression in song nuclei (*Figure 5i, j*). While the male read mapping of the W genes in modules F, G, and L does appear to shift in the song system relative to the surrounds, each of these modules contains a single W gene, and all three of these genes are lowly expressed (*Figure 5b*).

These results demonstrate that rather than being confounded by chromosome dose, WGCNA allowed us to resolve the effects of dose in an unbiased way. Module E grouped together the un-dosage-compensated Z chromosome and W chromosome genes across brain regions. In contrast, the Z chromosome genes placed in modules specialized for one or more song nuclei had some level of dosage compensation in males. This compensation appeared regionally patterned for Z chromosome genes in module G (enriched in two or more song nuclei depending on treatment), which were specialized beyond the normal chromosome dosage only within the song system. Average expression values of all sex chromosome genes for each brain region from vehicle-treated animals can be found in *Supplementary file 4* and *Supplementary file 5*.

## Candidate gene drivers of HVC specialization in E2-treated females

To reduce the 344 genes in module G to the putative drivers of HVC development, we next examined the relationship of membership in module G (correlation to MEG-G) to gene expression specialization in HVC at the level of single genes. We did this by testing for correlations between individual gene expression and the specialization of HVC in males (vehicle- and E2-treated) or E2-treated females

in each of the following four comparisons: (1) male HVC specialization relative to the surround; (2) E2-treated female HVC specialization relative to the surround; (3) male HVC relative to female HVC in vehicle-treated controls; and (4) E2-treated female HVC specialization relative to vehicle-treated female HVC. For all four comparisons, we found the higher the correlation of module G genes to the MEG-G, the higher the correlation with the vocal learning specialization (*Figure 6a–d*). This means that their expression was higher in male or E2-treated female HVC relative to the appropriate nonvocal learning controls across all comparisons. We identified genes of interest as being strongly correlated ($r^2 \geq 0.5$) to both the module G eigengene and vocal learning specialization (*Figure 6e–h*, higher magnification view of colored boxes in *Figure 6a–d*). All genes of interest exhibited a positive correlation with song nuclei gene expression specializations across all four comparisons (*Figure 6a–d*). We next generated a core gene list from module G that correlates with both E2- and sex-dependent HVC expression by intersecting these four vocal learning specialized gene sets (*Figure 6i*). We found a core set of 14 genes that strongly marked vocal learning-capable HVC in all comparisons and strongly correlated with the aggregate of module G expression (*Supplementary file 6*, *Figure 6—figure supplement 1*). The results of each individual comparison can be found in *Supplementary file 7*.

We examined the chromosomes of origin for these 14 core genes and found that three (*GHR*, *RGS7BP*, and *THBS4*) were on the Z sex chromosome (*Figure 6j*). This was a >3-fold significant enrichment over chance of Z chromosome transcripts (p=0.009, upper-tailed hypergeometric test) against a background of module-assigned genes. This result was statistically significant regardless of the background gene set, when using all genes (~21,000) or only module G members (344). These three Z chromosome module G genes in HVC not only exhibited >50% reduced expression in control female HVC relative to males, but also exhibited upregulated expression in male HVC relative to the surrounding DN regardless of treatment and upregulation in E2-treated female HVC relative to either the surround or vehicle-treated female HVC (*Figure 6—figure supplement 1*). In the context of our cross-region sex chromosome analysis (*Figure 3c–f*), only *RGS7BP* was sexually dimorphic outside of the song system, being absent in all female surround regions. Neither *GHR* (growth hormone receptor) nor *THBS4* was significantly depleted in any surround region comparison. Taken together, these results indicate that the three core genes in vocal learning-capable HVC on the Z chromosome are subject to additional E2-sensitive transcriptional regulation in HVC, separate from the Z chromosome transcript reduction seen throughout the female brain.

Of the 14 core genes total, several have been previously studied in the brains of other species which may inform their role in vocal learning systems. *THBS4* encodes a secreted extracellular matrix glycoprotein necessary for appropriate neuronal migration in the mouse (*Girard et al., 2014*) and is elevated sixfold in the human cortex compared with nonvocal learning primates (*Cáceres et al., 2007*). Human *EDA2R* was recently identified as a top correlate of cognitive performance and brain size *Harris et al., 2020*; it was also found in a human GWAS study that correlated it with circulating estrogen and testosterone levels (*Ruth et al., 2020*). Rare mutations in human *PHETA1* lead to Lowe oculocerebrorenal syndrome, which includes pathophysiology in seizures, mental retardation, and structural brain abnormalities (*Kornfeld et al., 1975*; *Ates et al., 2020*). *SIX2* is a homeobox domain-containing transcription factor that governs early brain and craniofacial development and provides neuroprotection from dopamine injury (*Garcez et al., 2014*; *Gao et al., 2016*). *GHR* encodes a transmembrane receptor whose activation controls cell division (*Dehkhoda et al., 2018*). The gene which encodes GHR's ligand, growth hormone (GH), is interestingly duplicated and undergoing accelerated evolution in the genomes of songbirds, is upregulated in the zebra finch auditory forebrain following the presentation of familiar song, and exerts anti-atrophic influence during chicken development (*Yuri et al., 2008*; *Rasband et al., 2023*; *Xie et al., 2010*). Based on these findings, we consider *GHR* as the most likely candidate gene related to the E2-sensitive atrophy of HVC in females.

## Discussion

The present study seeks to further our understanding of sexually dimorphic vocal learning in zebra finches by comparing the gene expression specializations in the song system between male and female birds at PHD30 and in response to E2 treatment, at the onset of atrophy of the female vocal circuit. The birds were given either a vehicle or E2 from hatching, which induces rudimentary vocal learning behavior in females where it would otherwise be absent. In control females, HVC appeared unspecialized at the level of gene module expression, with no significantly differentially expressed

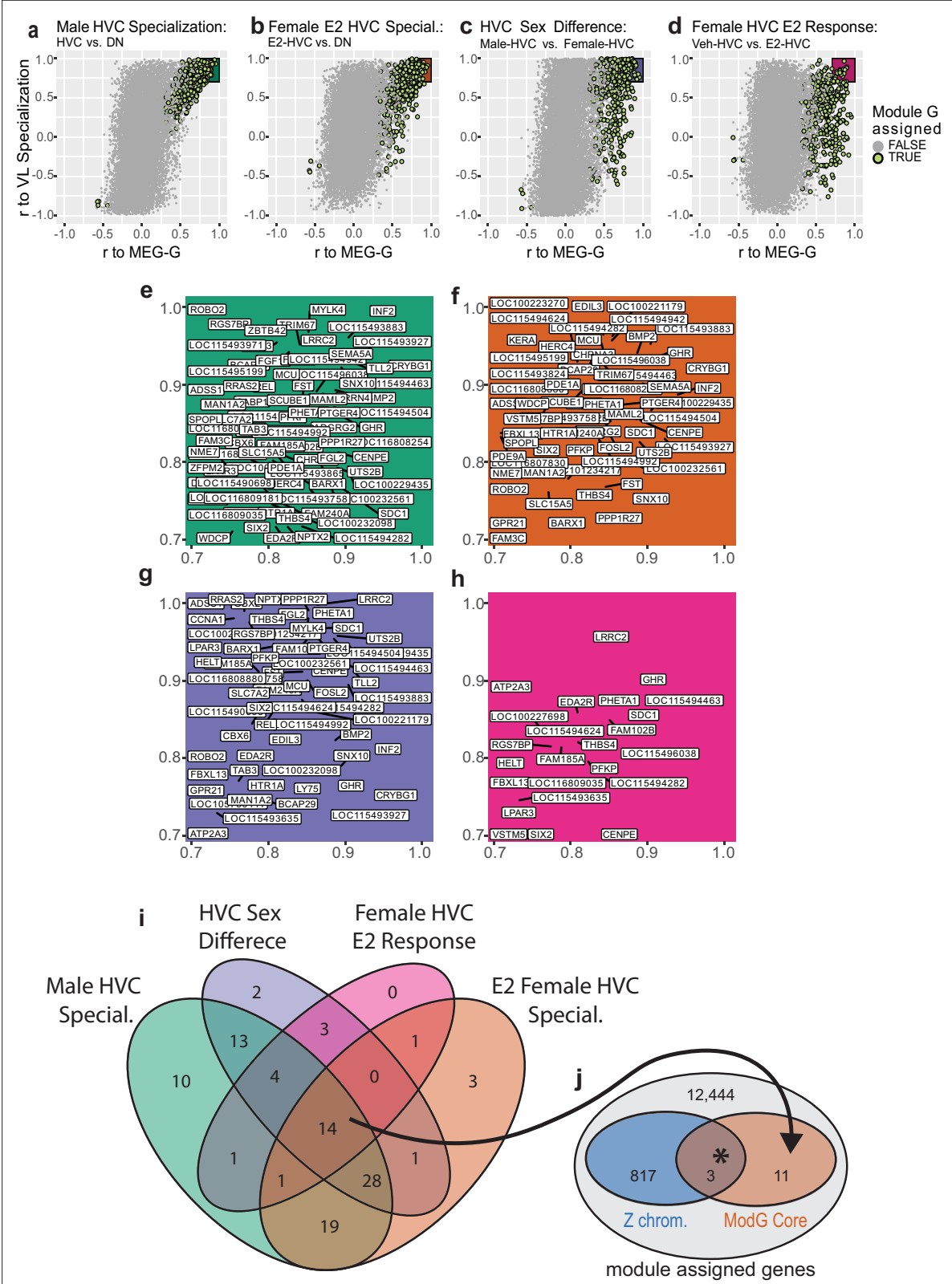

**Figure 6.** Identification of core genes in module G and their association to the Z chromosome. (**a–d**) Single gene continuous membership in module G (x-axis; Pearson's r to module eigengene [MEG] from module G) for all assigned genes vs correlation to vocal learning in masculine or masculinized HVC relative to samples from nonvocal learning females in each of the four comparisons: (**a**) male song system membership, comparing individual gene expression in male HVC samples of either treatment to expression in the surrounding dorsal nidopallium (DN); (**b**) female vocal learning capacity

*Figure 6 continued on next page*

*Figure 6 continued*

after E2, comparing E2-treated HVC to all other female DN or HVC samples; (**c**) sexual dimorphic gene expression within the song system, comparing vehicle-treated male and female song system components; (**d**) estradiol-responsive gene expression in female HVC, comparing E2-treated and vehicle-treated female HVC samples. Each point is a gene colored by module assignment, and the shaded area indicates gene of interest criteria for each comparison. (**e–h**) Blowup of shaded regions in **a–d**, respectively, showing genes of interest from each comparison. (**i**) Identification of core genes by intersecting the four gene sets of interest. (**j**) Enrichment of Z chromosome transcripts within the core genes. * indicates p=0.0087 by an upper-tailed hypergeometric test.

The online version of this article includes the following figure supplement(s) for figure 6:

**Figure supplement 1.** Expression of module G core genes in HVC and surrounding dorsal nidopallium.

MEGs compared to the surrounding nidopallium. However, in E2-treated females, HVC exhibited a subset of the observed male HVC gene expression specializations. Similarly, in the vehicle-treated females, the striatum located where Area X would also lack any specialized gene module expression, but the E2-treated female Area X had a subset of specialized gene expression as in males. This contrasts with RA and LMAN, which were similarly specialized in males and females in the absence of E2 treatment. Given that lesions of HVC prevent the emergence of Area X in E2-treated females (*Herrmann and Arnold, 1991*), these results support a model of zebra finch development where transcriptomic masculinization of female HVC by E2 is a critical event which facilitates the emergence of female Area X and ultimately endows these females to produce some rudimentary learned song.

How did E2 treatment produce the transcriptomic effects we observed with module G in female HVC, and what are the implications regarding sexually dimorphic vocal learning in the zebra finch? One possibility is that this process begins with increased estrogen receptor (ER) activity within developing HVC cells after being provided surplus activating ligand (*Frankl Vilches and Gahr, 2018*), followed by altered transcription of ER targets in the genome. Module G contained the AR (*Supplementary file 1*), which is believed to be a major downstream effector of the E2 response in female zebra finches (*Nordeen et al., 1986*). This initial transcriptional loading of the system would then have been processed by gene regulatory networks within each cell, spreading in effect through the transcriptome. It is possible that differential module G expression arose purely from traditional gene regulatory networks, where transcription factors form complex, elaborate feedback networks with themselves and the genes they regulate. However, this framework fails to explain why the core 14 vocal learning correlated genes from module G in HVC were enriched for transcripts from a single chromosome: the Z sex chromosome with halved copy number in females.

We hypothesize that the Z chromosome genes identified here are co-regulated, necessary components of a growth-enabling transcriptional program downstream of ER, represented by module G. This module G transcriptional program could be specialized to developing male HVC by the expression of patterning genes, such as *SIX2*, early in development and maintained through persistent *GHR* signaling. We propose that these Z chromosome transcripts in module G are reduced in females by lower haplotype dosage during development and thus fail to specialize female HVC. Due to the lower abundance of gene products from module G, female HVC may be unable to accommodate new neurons during juvenile development and fail to facilitate the emergence of its downstream target Area X. Similarly, without a sufficiently developed HVC, RA lacks one of its major inputs and may subsequently atrophy. We propose that E2 masculinizes female song behavior by increasing the abundance of these module G transcripts in HVC, increasing specialized HVC growth, and facilitating the emergence of HVC's other major target, Area X (*Figure 7*). This model of sex chromosome-influenced song system development is consistent with recent work comparing male and female zebra finch transcriptomes from RA at young juvenile (PHD20) and young adult (PHD50) ages in un-manipulated birds (*Friedrich et al., 2022*). While that study proposed that the role of the sex chromosome in maintaining transcriptomic sex differences diminishes across development as the proportion of specialized genes that originate on the sex chromosomes diminishes, this effect was driven by large increases in differentially expressed autosomal genes rather than by any reduction in sex chromosome dimorphism; the percentage of differentially expressed Z chromosome genes increased from 28% at PHD20 to 39% at PHD50 (*Friedrich et al., 2022*). This leads us to conclude that sexually dimorphic Z chromosome expression in juveniles precedes the sexually dimorphic expression of the autosomes seen in adults. This is consistent with our hypothesis that sufficient expression of select Z chromosome gene products (GHR, etc.) is necessary for subsequent autosomal song system specializations (module

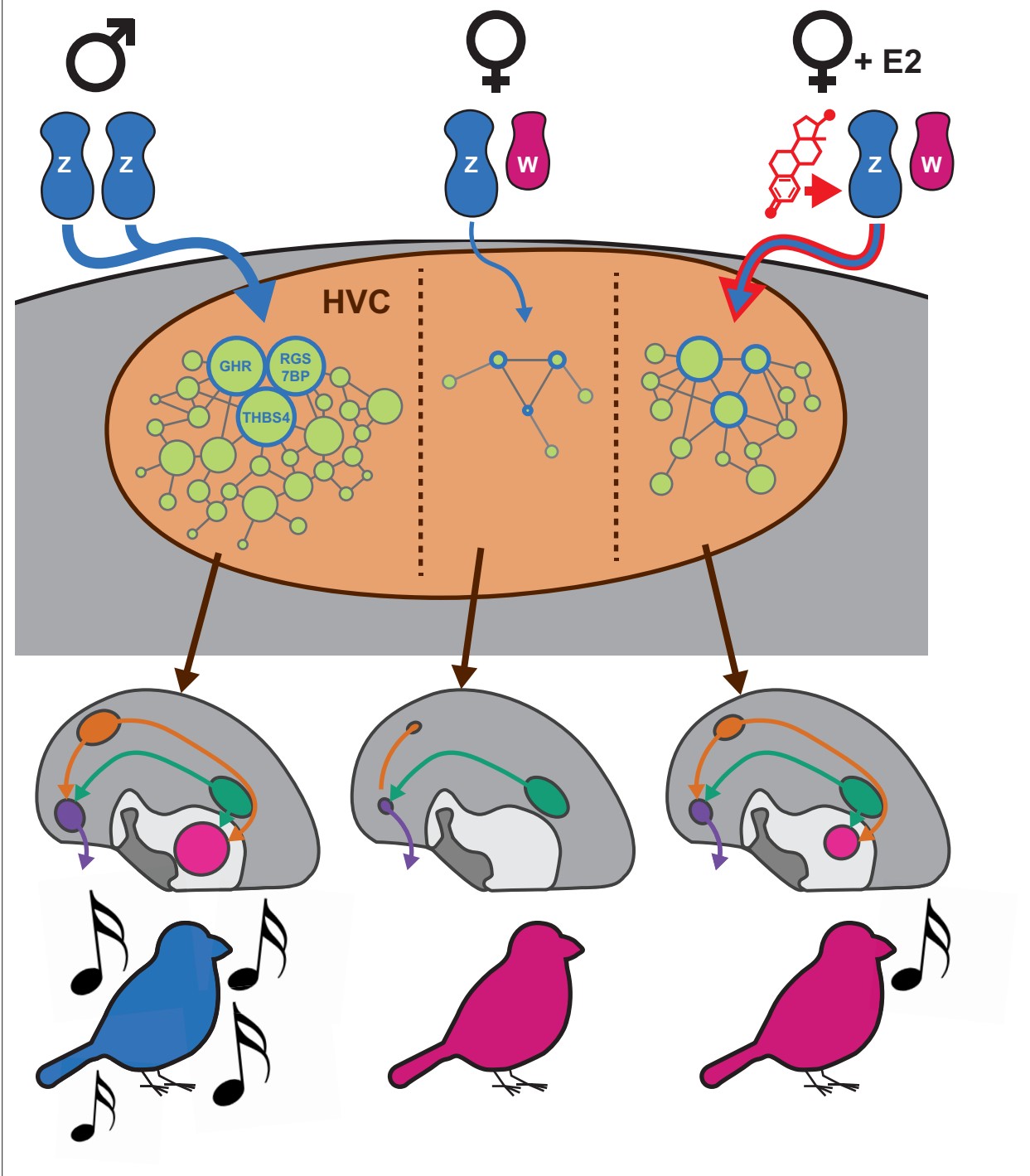

**Figure 7.** Proposed model of sexually dimorphic zebra finch vocal learning. We propose that estradiol treatment in female zebra finches masculinizes song behavior by overcoming insufficient Z sex chromosome dosage in HVC to increase the expression of transcripts normally depleted in females. The Z chromosome genes upregulated by E2 are components in a larger proliferative genetic program which prevents HVC atrophy in males and allows for its expansion late in development. The upregulation of these genes allows for the increased specialization of the gene networks they participate in, promoting HVC development sufficiently to enable rudimentary vocal learning in females.

G). Further, our results are consistent with GH's known role in avian neuroprotection, with elevated signaling associated with the survival of chicken neurons during rounds of pruning in the developing retina (*Harvey et al., 2009*).

Our results help refine the traditional notion that hormonal signaling organizes the brain to produce sexually dimorphic behaviors independent of neuronal sex chromosome content (*Phoenix et al., 1959*; *Arnold, 2009*). Instead, our data indicate that sexual dimorphism of the zebra finch song learning ability was likely established by the interaction of sex hormone signaling and sex chromosome gene expression within HVC during development. These findings are thematically similar to work in the Four Core Genotypes mouse model (*De Vries et al., 2002*), where chromosomal and gonadal sex are separable by translocating the sex-determining *Sry* gene. In these mice, sex chromosome composition regulates sexually dimorphic brain gene expression, circuit anatomy, and behaviors (*De Vries et al., 2002*; *Chen et al., 2009*; *McPhie Lalmansingh et al., 2008*; *Cox and Rissman, 2011*), though the sex chromosome genes responsible remain unknown. Additional experiments manipulating the candidate genes implicated here in developing HVC to both mimic and prevent the action of E2 in female zebra finches are needed to test these hypotheses.

In addition to these vocal learning-focused results, performing unbiased gene network analysis with samples taken throughout the zebra finch telencephalon in both sexes revealed a surprisingly strong relationship between the modular organization of telencephalic gene expression and the chromosomal structure of the genome. Each of the 12 modules identified by WGCNA was enriched for genes from at least one chromosome. Of these significant enrichments, those on the micro- and sex chromosomes stood out as particularly strong. Module E encompassed the majority of W and Z chromosome genes and was sexually dimorphic in its expression in all sampled regions. Indeed, we found that roughly ⅓ of sex chromosome transcripts are significantly sexually dimorphic in all nonvocal brain regions, and roughly ⅔ are sexually dimorphic in at least one region. This half-brain-wide and half-regionally patterned sexual dimorphism observed from the sex chromosomes may act as a substrate for the evolution of sexually dimorphic behaviors generally, with brain-wide shifts in expression providing a transcriptional background for the evolution of sex-specific patterning in additional sex chromosome genes. These regionally patterned sex chromosome genes could then recruit gene expression networks from somatic chromosomes, resulting in the sex-specific anatomical patterning of gene expression from throughout the genome. This general model is consistent with local reduction of GHR and other Z chromosome transcripts in the developing HVC lineage, leading to a loss of specialized module G expression in female HVC from somatic chromosomes. This hypothesis of an interaction between sex chromosomes and autosomes linked to the gain and loss of vocal learning in one sex can be tested in future studies.

## Materials and methods
### Animal handling and sample preparation

The 96 samples used in the present analysis are the E2- or vehicle-treated subset of a previously published RNA-seq dataset (*Choe et al., 2021*). We briefly redescribe our methodology here. All animal procedures were approved by the IACUC of Duke University.

E2 (Sigma E1024-1G) was dissolved in DMSO (100 mg/mL) and then diluted in olive oil (1 mg/mL). 30–50 µL of E2 sample or DMSO-only vehicle was applied to the flank of male and female zebra finches daily from PHD0 to PHD14 and on alternating days from PHD15 to PHD30 (n=3 per sex-treatment combination). We have previously shown that this treatment program is sufficient to induce song system masculinization in E2-treated female zebra finches (*Choe et al., 2021*).

On PHD30, animals were sacrificed following 1 hr of dark isolation. Animals were anesthetized by isoflurane inhalation and rapidly decapitated. Brain hemispheres were dissected, embedded in OCT, and flash-frozen in an ethanol and dry ice slurry. Sections were taken from the right hemisphere coronally at 14 µm onto polyethylene naphthalate (PEN) membrane slides for RNA isolation and adjacent sections taken on charged glass slides for histology or in situ hybridizations. From the PEN membrane slides, song nuclei and surrounding control regions were laser capture microdissected (LCM) using an ArcturusXT LCM system (Nikon) guided by a Nissl-stained tissue series for each animal. No sample pooling was performed; each sample originated from a single bird. This is eight samples worth of

RNA-seq data per bird for 12 birds, providing three samples per sex-treatment-region combination, roughly a terabyte of read data.

RNA was extracted from the LCM isolated tissue samples using the Arcturus Picopure Kit (Applied Biosystems KIT0204) following the manufacturer's instructions. RNA quality was assessed using an Agilent 2100 Bioanalyzer and the RNA 6000 Pico Kit (Agilent 5067-1513). Next, cDNA was synthesized using the SMART-Seq v4 Ultra Low Input RNA Kit (Takara 634892). Sequencing libraries were made with the NEBNext Ultra II DNA Library Prep Kit (New England Biolabs E7645L) and cleaned up using SPRIselect beads (Beckman Coulter B23317). Libraries were sequenced by Novogene Co., Ltd. on the NovaSeq 6000 platform (Illumina) and S4 flow cells resulting in 150 bp paired-end reads.

## RNA-seq read mapping and quality control

RNA-seq reads were first trimmed to remove adapters and low-quality base calls using Trimmomatic (*Bolger et al., 2014*) and then mapped to a high-quality VGP female zebra finch nuclear genome ( bTaeGut2.pat.W.v2, GCF_008822105.2) (*Rhie et al., 2021*) using STAR (v2.7.1) (*Dobin et al., 2013*). Uniquely mapped reads were then tallied at the level of genes using Rsubread::featureCounts (R-3.6.3) and then counts normalized to fragments per kilobase of transcript per million mapped reads (*Liao et al., 2014*). Multi-mapped reads were rejected as they have a higher probability of being associated with technical artifacts of sequencing or genome assembly. Read-based quality control was performed with FastQC (Babraham Bioinformatics) with reports prepared by MultiQC (Python 3.5.5) (*Ewels et al., 2016*). This workflow was automated by the CountMatrix pipeline (https://github.com/ mattisabrat/CountMatrix/ copy archived at *Davenport, 2021*). We next removed two outlier samples (one male vehicle HVC and one female vehicle RA) based upon hierarchical clustering of the sample space before computing gene-to-gene correlations (*Figure 1—figure supplement 1*).

## Gene module identification

All remaining analyses were completed in R 4.2 unless otherwise specified. Data was wrangled in the tidyverse, and custom visualizations produced with ggplot, ggdendro, VennDiagram, RColorBrewer, and ggpubr (*Wickham et al., 2019*; *Wickham, 2009*). Unsigned topological overlaps between genes (gene-to-gene correlations) were calculated in a single block with WGCNA::blockwiseModules with a soft-thresholding power ($\beta$) of 6 based on scale-free topology fit and model connectedness as described in the WGCNA vignette (*Figure 1—figure supplement 2*; *Langfelder and Horvath, 2008*). Next, we determined an appropriate WGCNA parameterization quantitatively and qualitatively by sweeping the module size and tree cut height parameters of WGCNA::recutBlockwiseTrees. We selected our model for analysis by examining the resulting gene assignments and sample-sample distance matrices. We were able to increase or decrease the proportion of genes assigned to WGCNA modules by lowering or raising the minimum module size parameter, respectively. We found that setting the minimum size parameter below 100 genes included more genes, but with models that increasingly overfit single samples, producing obvious outliers in the distance matrix (*Figure 1— figure supplement 3*). Correspondingly, raising the minimum size parameter beyond 100 included fewer genes, but did not greatly reduce the number of outlier samples. Based on this, we selected 100 as the minimum module size, parameterizing to explain as much transcriptomic variance as possible while minimizing the number of technically overfit samples (*Figure 1—figure supplements 2–4*).

## Module association to vocal learning

MEGs from each module were correlated against binarized song system membership, vocal learning capability, or sex, and the statistical significance of each correlation was assessed using WGCNA::corPvalueStudent. This was done in the following sample subsets by node: male samples broken out by treatment; female samples broken out by treatment; all female samples; song system components from either sex treated with vehicle; and surrounding control regions from either sex treated with vehicle. Within each of the four sex-treatment combinations, we compared the song system components to surrounds at each node. Within all female samples, we compared the vocal learning-capable E2-treated song system elements to all other female samples from each node. Within the vehicle song systems and vehicle surrounds, we compared male and female finches for each region.

## Module GO and convergent vocal learning gene expression signature enrichment

Module-assigned zebra finch genes were mapped to their 1:1 human orthologs where possible, dropping unmapped or multi-mapped genes, using orthofindR::getOrthos (https://github.com/ggedman/orthofindR; *Gedman, 2019*) which wraps Ensembl's biomaRt. Uncorrected p-values for the enrichment of human GO terms within the human orthologs of module G were calculated using generally applicable gene set enrichment (GAGE) implemented in gage::gage (*Luo et al., 2009*). To determine if the genes previously shown as convergently differentially expressed in the zebra finch song system and human speech brain regions mapped to specific modules, we treated these five gene lists identically to GO terms and tested for their significant enrichment across the human orthologs of each module also using GAGE.

## Analysis of sex chromosome gene expression independent of vocal learning or E2 treatment

Sex chromosome transcripts, regardless of WGCNA module assignment, were examined in the vehicle-treated nonvocal learning producing surround samples for each node. To find the most consistently expressed and depleted W and Z chromosome genes, respectively, we correlated expression of each sex chromosome transcript with sexual dimorphism within each region, such that expressed W genes would be positively correlated and depleted Z chromosome genes would be anticorrelated. We computed correlations and p-values using the WGCNA corAndP function; upper-tailed for the W chromosome and lower-tailed for the Z chromosome. Genes significantly expressed or depleted across regions were then intersected to identify consistently regulated transcripts across the telencephalon.

## Module enrichment on chromosomes

To associate modules to chromosomes, we bootstrapped FDR-corrected p-values for the enrichment of each chromosome-module pairing by randomizing the mapping of genes to modules 50k times and calculating the fold-enrichments observed on each chromosome from each module in each randomization to empirically determine the null distributions. The calculation of bootstraps and p-values was performed in Python 3.5.5 and parallelized using joblib's Parallel.

## Identification of core module G genes in HVC and Z chromosome enrichment

We defined genes of interest as the subset of significantly (t-test for correlation: $p \leq 0.05$) vocal learning capability correlated genes in HVC whose expression correlated to module eigengene G (MEG-G) across the dataset at $r^2 \geq 0.5$ and to vocal learning at $r^2 \geq 0.5$ in at least one of the four vocal learning comparisons in HVC, calculated using WGCNA::corAndP. These comparisons were: all male HVC samples against all male DN samples; female E2-treated HVC against all other female samples at the node, including vehicle-treated HVC; vehicle-treated male HVC against vehicle-treated female HVC; and E2-treated female HVC against vehicle-treated female HVC. We defined core genes as those meeting this criteria for all four HVC vocal learning comparisons. We tested the statistical significance of Z chromosome enrichment in this core gene list with an upper-tailed hypergeometric test, implemented in phyper, where each core gene is a sampling event without replacement from module-assigned genes.

## Acknowledgements

Funding for this work was provided by the Howard Hughes Medical Institute (EDJ), NIH-NIDCD R01-DC016224 (HM), and the NSF-GRFP (MHD). We thank Gregory Gedman, Giulio Formenti, Caitlin Gilbert, Lindsey Cantin, César Vargas, Chul Lee, and Jason Manley for conversations during visualization and analysis. We also thank Alipasha Vaziri and Tobias Nöbauer for providing the computing infrastructure used throughout. This analysis is a memorial to Mark Konishi (1933–2020) whose work on this topic influenced us greatly.

# Additional information

### Competing interests

Hiroaki Matsunami: HM has received royalties from Chemcom, received research grants from Givaudan and received consultant fees from Kao.

### Funding

| Funder | Grant reference number | Author |
| --- | --- | --- |
| National Science Foundation | NSF-GRFP | Matthew Davenport |
| Howard Hughes Medical Institute | | Erich Jarvis |
| National Institute on Deafness and Other Communication Disorders | R01-DC016224 | Hiroaki Matsunami |

The funders had no role in study design, data collection and interpretation, or the decision to submit the work for publication.

### Author contributions

Matthew Davenport, Conceptualization, Resources, Data curation, Software, Formal analysis, Validation, Investigation, Visualization, Methodology, Writing – original draft, Project administration, Writing – review and editing; Ha Na Choe, Conceptualization, Data curation, Investigation, Methodology, Writing – review and editing; Hiroaki Matsunami, Conceptualization, Supervision, Project administration; Erich Jarvis, Conceptualization, Supervision, Funding acquisition, Project administration, Writing – review and editing

### Author ORCIDs

Matthew Davenport ⓘ https://orcid.org/0000-0002-5699-6295
Ha Na Choe ⓘ http://orcid.org/0000-0001-9535-5258
Hiroaki Matsunami ⓘ http://orcid.org/0000-0002-8850-2608
Erich Jarvis ⓘ https://orcid.org/0000-0001-8931-5049

### Ethics

The 96 samples used in the present analysis are the E2 or vehicle treated subset of a previously published RNAseq dataset. All animal procedures were approved by the IACUC of Duke University.

Reviewer #3 (Public review): https://doi.org/10.7554/eLife.89425.3.sa1
Author response https://doi.org/10.7554/eLife.89425.3.sa2

# Additional files

### Supplementary files

Supplementary file 1. Gene module assignments.

Supplementary file 2. Gene Ontology (GO) enrichments by module.

Supplementary file 3. Sexually dimorphic sex chromosome transcripts across regions.

Supplementary file 4. Z chromosome expression ratios.

Supplementary file 5. W chromosome expression ratios.

Supplementary file 6. Module G core VL genes.

Supplementary file 7. Sex- and E2-dependent HVC specializations.

MDAR checklist

## Data availability

All raw data for this experiment is available on the NCBI Sequence Read Archive (accession: PRJNA698257). The count matrix, quality control results, and analysis code is available online (https://github.com/mattisabrat/sex-and-song copy archived at *Davenport, 2023*).

The following dataset was generated:

| Author(s) | Year | Dataset title | Dataset URL | Database and Identifier |
|---|---|---|---|---|
| Choe HN, Davenport MH, Jarvis ED, Matsunami H | 2021 | Brain transcriptome from estrogen manipulated juvenile zebra finches | https://www.ncbi.nlm.nih.gov/bioproject/PRJNA698257 | NCBI BioProject, PRJNA698257 |

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
