## [Editor Report · eLife Assessment]

This study is **valuable** as it provides information about the genes regulated by sex hormone treatment in song nuclei and other brain regions and suggests candidate genes that might induce sexual dimorphism in the zebra finch brain. The analysis presented is thorough and detailed. Whereas the evidence for gene regulation by hormone treatment is well supported, the evidence for an association of those genes with song learning (as written in the title) is **incompletely** supported as no manipulation of song learning or song analysis was conducted.

---

## [Referee Report · Reviewer #3 (Public review)]

Summary:

Davenport et al have investigated how a masculinizing dose of estrogen changes the transcriptomes of several key song nuclei song and adjacent brain areas in juvenile zebra finches of both sexes. Only male zebra finches sing, learn song, and normally have a fully developed song control circuitry, so the study was aimed at further understanding how genetic and hormonal factors contribute to the dimorphism in song behavior and related brain circuitry in this species. Using WGCNA and follow-up correlations to re-analyze published transcriptome datasets, the authors provide evidence that the main variance of several identified gene co-expression modules significantly correlates with one or some of the factors examined, including sex, estrogen treatment, regional neuroanatomy, chromosomal placement, or vocal learning, noting that the latter is largely based on inference due to expression in song control nuclei.

Strengths:

Among the main strengths are the thorough gene co-expression module and correlation analyses, and the inclusion of both song nuclei and adjacent areas, the latter serving as sort of controls for areas that are not dimorphic and likely broadly present in birds in general. In situ hybridization data discussed in a previous publication (Choe et al., Hormones and Behavior, 2021) provides some support for the neuroanatomical specializations of gene expression. It is also significant that the transcriptome re-analysis was performed with an improved genome assembly that also includes the sex chromosomes, thus expanding the Z/W chromosome gene analyses in Friedrich et al, Cell Reports, 2022. The most relevant finding is arguably the identification of some modules where gene expression variation within song nuclei correlates with hormonal effects and/or gene location on sex chromosomes, which are present at different dosages between sexes. Sex differences in gene expression in areas that are not song nuclei may also bring insights into functions other than song behavior or vocal learning. The study also shows how a published RNA-seq dataset can be reanalyzed in novel and informative ways.

Weaknesses:

The validation of the inferred direction of regulation in the identified co-expression modules is limited to the in situ data mentioned above. Further evidence that representative genes in the main modules differ in expression when comparing sexes or E2- vs VEH-treated tissues using independent samples and/or methods would provide further validation and enhance rigor. Most importantly, E2 is known to exert various actions on brain physiology and neuronal function. Because there was no manipulation of candidate genes, nor assessment/manipulation of vocal behavior or vocal learning, an involvement of the identified candidate genes in setting up the sexual dimorphism of the song system or song behavior was not directly tested in this study. For the latter reason, the implication of the Title (..."gene expression associated with vocal learning...") is not well supported. While novel insights were gained into brain expression of Z chromosome genes, it cannot be excluded that the higher male expression of some Z genes may not affect brain cell function and thus may not require active compensation (as discussed for nucleus RA in Friedrich et al, Cell Reports, 2022).

---

## [Author Response]

The following is the authors’ response to the original reviews.

**eLife Assessment**
This study is useful as it provides further analysis of previously published data to address which specific genes are part of the masculinizing actions of E2 on female zebra finches, and where these key genes are expressed in the brain. However the data supporting the conclusion of masculinizing the song system are incomplete as the current manuscript is a re-analysis of differential gene expression modulated by E2 treatment between male/female zebra finches without manipulation of gene expression. The conclusions (and title) regarding song learning are also incompletely supported with no gene manipulation or song analysis. Importantly, the use of WGCNA for a question of sex-chromosome expression in species without dosage compensation is considered inadequate. As the experimental design did not include groups to directly test for song learning, and there was also no analysis of song performance, these data were also considered inadequate in that regard.

We are sorry the editor felt the manuscript so incomplete and inadequate. Though the tone of this assessment seems more severe than the below reviewer comments, we are also happy to see that the editor has considered our paper further for a revised publication, based on the reviewer’s comments. We address the editor’s comments as follows:

While we agree that manipulation of some of the genes we discovered, whose expression levels are E2-sensitive in the song system, would take the study further in validating some proposed hypothesis in the discussion of the paper, we don’t think the outcome of gene manipulations would change the major conclusions from the results of the paper. In this study we performed estrogen hormone manipulations, with causal consequences on gene expression in song nuclei and associated song behavior. In a way this is analogous to gene manipulations, but manipulating directly the action of estrogen. The categories of genes impacted, and the differences among the sex chromosomes wouldn’t change.

For the comment on WGCNA being inadequate for addressing questions on sex chromosome expression in species without dosage compensation, we think the evidence in our data does not bear that out. One main result of this paper is the separation of Z chromosome transcripts whose expression is most strongly regulated by chromosomal dosage (WGCNA module E) across regions from those subject to additional sources of regulation in song nuclei (other modules). It seems to us that rather than being confounded by the lack of dosage compensation, WGCNA allowed us to better resolve the effects of dosage on different genes within the sex chromosomes. We have added a new figure more directly examining sex chromosome transcript abundance within different modules. Briefly, we found that module E assigned Z chromosome genes exhibited almost exactly the male-biased expression ratio expected from no dosage compensation while the Z chromosome genes in song nuclei assigned to other modules were expressed below the dosage predicted value, consistent with module E containing those genes whose expression are most strongly regulated by dose across all brain regions sampled.

At its core, WGCNA finds sets of correlated genes. The biological reality of the zebra finch transcriptome is that Z chromosome expression is largely anti-correlated with W chromosome due to dosage. However, this dosage effect is not felt equally by all genes and WGCNA provides an unbiased computational framework which can be used to separate dose from other potential sources of gene regulation. This is why roughly ⅓ of Z chromosome genes are not assigned to module E; for example the growth hormone receptor is assigned to module G based on its correlation with genes upregulated within HVC.

“As the experimental design did not include groups to directly test for song learning, and there was also no analysis of song performance, these data were also considered inadequate in that regard.”

Concerning the comment on no analysis on song performance in the paper, all such analyses were conducted on our previous study on the same animals (Choe et al. 2021, Hormones & Behavior). The birds considered here were sacrificed at PHD30, prior to the onset of learned song behavior. However, females treated with E2 the same at the same time and allowed to mature into adulthood, went onto to develop rudimentary song. Further, induction of rudimentary song learning in females following E2 treatment has been well established since the early ‘80s. We have added the following text toward the end of the intro to make this more clear:

“While the birds for this study were sacrificed prior to the developmental presentation of song behavior, we have previously shown that female finches treated in exactly the say way with E2 go on to produce rudimentary imitative songs as adults (Choe et al 2021), consistent with the known induction of vocal learning in females by E2 (REF).”

**Reviewer #1 (Recommendations For The Authors):**
Overall, this is a wonderfully designed and executed study that takes full advantage of new resources, such as the most complete zebra finch genome assembly yet, as well as the latest methods. I have very few suggestions as to the improvement of the manuscript. They are as follows:Results Section:In the paragraph "Identification of gene expression modules in song nuclei":"The E2-treated females in this study had similarly sized song system nuclei as males, indicating that E2 treatment prevented atrophy."Clarify if this comparison is to treated and/or untreated males.

We thank the reviewer for their comment. The relative differences in the song nuclei sizes between the E2-treated females and the other groups is more complex that our original sentence implied. We have revised the main the text as follows

“In our previous study, we found that estradiol treatment in PHD30 females caused HVC to enlarge and Area X to appear when it normally does not develop in females, but both at sizes less than in untreated or treated males.The sizes of PHD30 female LMAN RA were already the sizes as seen in males, as the later has not atrophied yet at this age(25).”

In the paragraph "Sex- and micro-chromosome gene expression across the telencephalon": "These animal and chromosome specific shifts in the transcriptomes could represent the systemic effects of allelic chromosomal structural variation..."The authors should clarify the meaning of a"llelic chromosomal structural variation" in this context, as it is an unusual phrase. Major chromosomal structural variation seems unlikely to produce these effects. Is it also possible that animal-specific modules with brain-wide higher could also result from laboratory contamination between all samples from one animal? This is not too likely but perhaps should be acknowledged or ruled out.

We have removed the word allelic, which was unnecessary. We can’t envision how laboratory contamination could occur such that all of one animal’s samples would be affected to produce the observed result which is module and chromosome specific. An animal wide effect could emerge during sacrifice, but we can think of no reason that would affect these modules and not others. Rather, the most likely explanation is biological natural difference between animals. We have added this consideration of alternative explanations.

In the section "Candidate gene drivers of HVC specialization in E2-treated females":When discussing GHR's role in cell growth and proliferation, the authors' argument could be expanded by including the documented role of GH signaling in anti-apoptotic protection of neurons from rounds of neural pruning during development as documented in the chicken, e.g. • Harvey S, Baudet M-L, Sanders EJ. 2009. Growth Hormone-induced Neuroprotection in the Neural Retina during Chick Embryogenesis. Annals of the New York Academy of Sciences, 1163: 414-416. https://doi.org/10.1111/j.1749-6632.2008.03641.x

We thank the reviewer for sharing this publication with us.. We have added the following sentence to our discussion with the above citation. “Further, our results are consistent with growth hormone’s known role in avian anti-apoptotic protection, with elevated signaling associated with the survival of chicken neurons during rounds of pruning in the developing

retina.”

The authors' argument of the relevance of the passerine GH duplication would be strengthened by citing:• Rasband SA, Bolton PE, Fang Q, Johnson PLF, Braun MJ. 2023. Evolution of the Growth Hormone Gene Duplication in Passerine Birds, Genome Biol Evol, 15(3) https://doi.org/10.1093/gbe/evad033. Greatly expands on the Yuri et al. paper cited by characterizing of the molecular evolution of these genes across hundreds of avian species, supporting positive selection on multiple amino acid sites identified in both ancestral and duplicate (passerine) growth hormone.• Xie F, London SE, Southey BR et al. 2010. The zebra finch neuropeptidome: prediction, detection and expression. BMC Biol 8, 28. https://doi.org/10.1186/1741-7007-8-28 The authors report significantly different expression of the ancestral GH gene in the adult male zebra finch auditory forebrain after different song exposure experiences.

We have amended the results section sentence and added all suggested citations. The sentence now reads: “The gene which encodes growth hormone receptor’s ligand, growth hormone, is interestingly duplicated and undergoing accelerated evolution in the genomes of songbirds (Rasband et al 2023); the GH ligand has been found to be upregulated in the zebra finch auditory forebrain following the presentation of familiar song (Xie et al 2010).”

Figures:- Figure 1B. "Duration of sex typing" being a shorter bar compared to the others is not fully explained in the experimental design. Presumably at the end of this time period, the sex is non-invasively, phenotypically evident. I suggest an arrow pointing to the PHD/PHD range when sex is apparent in plumage/anatomy.- Figure 4. Caption appears to be truncated; "across all... genes"?

Fixed

- Figure 5. For 5E, 5F, 5G, 5H, consider enlarging the plots so overlapping gene symbols are readable. Alternately, smaller numbers or symbols could be used with a key in areas where overlapping symbols are hard to prevent.

We agree that these are not the easiest to read; we originally offset the symbols in R to minimize overlaps, but it can only do so much for the more crammed panels. We have now added a supplemental .xlsx file with the underlying data from each of the 4 tests for readers that want to examine the data in more detail.

**Reviewer #2 (Recommendations For The Authors):**
Since WGCNA methods will inherently draw together sex-chromosome genes into the same module in systems without dosage compensation, I suggest the authors rerun the WGCNA using only female samples and only male samples. Then identify the composition of modules that differ between E2 and vehicle-treated females and compare these genes to males. Then from male WGCNA identify the composition of modules that differ between E2 and vehicle-treated males and compare to female modules.

We thank the reviewer for their suggestions. However, we believe it is not as strong as the approach we used, which is grouping data from both sexes in the WGCNA analyses in a study that is looking for sex differences. The reviewer's proposed approach amounts to computing modules twice (once per sex), determining song system specialized modules and E2 responsive modules in both settings, then intersecting the two sets to find corresponding modules, all done to prevent the non-dose compensated sex chromosome genes from being drawn into the same module.

While WGCNA does group the majority of sex chromosome genes into module E, it does not categorize them all this way (Fig 3). The module classification instead differentiates those sex chromosome genes whose expression are most explained by chromosome dosage / sex across regions (modE) from those whose expression is controlled by other sources of regulation; for an example of the latter, the growth hormone receptor (GHR) is one of several Z chromosome genes classified into modG as its expression better correlates with the genes specialized to HVC than it does with the majority of dosage-dependent Z chromosome genes found in modE. Further, to remove biological sex as a variable in a WGCNA analysis that is focused on sex differences seems counterintuitive.

Instead, to quantitatively address the reviewer’s concern, we conducted additional analyses, that led to an added new figure, associated text, and tables, that better describes sex/chromosome dosage effects on the abundance (FPKM) and expression ratios of sex chromosome transcripts by module irrespective of brain region (Fig. 5). We find that the Z chromosome genes in modE were expressed at the expected chromosome dosage in the non-vocal surrounding regions (65.06% observed vs 66.6% expected) while in other modules, other Z chromosome genes were expressed at intermediate levels between equal expression and the expected chromosomal dosage. For example, the Z chromosome content of modules D and H exhibited near equal expression between sexes. Within the song system, Z chromosome gene content of modG was highly expressed in males beyond what is expected from chromosome dosage, consistent with modG’s male-specific upregulation in song nuclei relative to surrounds in the absence of E2. These results better demonstrate that in our WGCNA on the combined dataset we are able to separate those Z chromosome genes whose expression is predominantly dosage controlled from those subject to additional regulation such as song system specialization.

Fig. S3 Legend: 'Black arrow' -> 'Red arrow'

Change made.

Fig. S5 - What part of the figure shows the 'human convergent signature'? Also, simply listing the number of genes mapped to a chromosome is misleading to readers unfamiliar with the zebra finch genome, you should either provide the number of genes on each chromosome or present as corrected by that number.

Fig. S5 was the same type of analyses in Fig. 3 but with an older zebra finch genome assembly, where we had not included the panel a for enrichments with genes convergent in expression between songbird song regions and humans speech brain regions. However, we see that Fig. S5 was not adding any new important information to the paper, so we removed it.

For the chromosome analyses in Fig. 3b, we provide both the total raw number of module assigned genes broken down by chromosome (The black bar plots on the right) as well as a statistical fold-enrichment value of modules per chromosome. Given the number of genes per chromosome and genes per module in our data, we computed the fold-enrichment for each intersection (observed intersection size / expected intersection size). To test for the significance of these enrichments, we bootstrapped FDR corrected p values for the enrichment of each chromosome-module pairing by randomizing the mapping of genes to modules to construct a null distribution of fold enrichments for each intersection. Our intent was not to describe the size of the chromosomes themselves, information readily available elsewhere, but to show the disproportionate chromosomal origins of the gene sets considered by this study. Performing this enrichment test using all annotated genes per chromosome would artificially increase enrichment values and make the analysis less conservative by confounding the results with the inherent enrichment for “brain function” in the assigned genes relative to all genes.

At several places you say "we correlated expression of each sex chromosome transcript with sexual dimorphism within each region, such that expressed W genes would be positively correlated and depleted Z chromosome genes would be anticorrelated." What was the sexual dimorphism that was being correlated with? Is this the eigengene?

We thank you for this comment. Our language was less clear than it could be. We tested for correlations of both the eigengene and the individual gene expression profiles with the biological sex of the animals. We have changed the text to:

“To do this, we tested for a correlation between the expression of each sex chromosome transcript to the animals’ sex within each brain region. We found that female-enriched transcripts were positively correlated with sex and male-enriched transcripts were anticorrelated (Fig. 4f,g).”

Fig. 4A: The 'true/false' boxes and animal A-L is confusing and unnecessary. I'd suggest just using M and F (or sex symbols) with a horizontal line below each set of 3 for respective E2 and Veh.

Change made.

**Reviewer #3 (Recommendations For The Authors):**
General comments:After the initial characterization of the datasets and module identification, it is quite hard to follow the logic of the data presentation in the various other Results sections or to clearly understand how they relate to the main stated goal to identify factors related to sex differences in vocal learning. The most relevant findings relate to the presumed actions of hormone treatment and sex chromosome gene dosage in song nuclei, whereas analyses of other brain areas, other chromosomes, or speech-related genes serve more as controls and/or appear as distractions from the main theme. A suggestion to increase the clarity of the presentation and potential impact of the study is to change the order of the presentation, focusing first on the specific analyses and comparisons that most directly speak to the main goals of the study, and then secondarily and more briefly presenting the controls or less related comparisons.

The reviewer’s suggestion for the results section organization is exactly what we had tried to do. We opened the first paragraph on identification of modules, then presented the song nuclei specific modules, followed by E2-changes to those modules; and the followed by other specific results for the remainder of the paper, including module enrichments to specific chromosomes. The reviewer mentioned our analyses of “other brain areas” (which we assume to mean the non-vocal surround regions), other chromosomes (which we assume means autosomes) and speech-related genes as controls were a distraction in the paper; but within our analysis, these other brain regions are essential controls needed to assess the song-system specificity of any observed sex differences observed from the very first paragraphs of the results; the autosomes were not controls for sex chromosome results, but primary results in of themselves; the overlap with speech-related genes was also not a control, but a novel discovery. We have revised these points in the paper to make them clearer, and revised some of the section titles and transitions between sections to help increase clarity of the main storyline of the paper.

A related comment is that many of the inferences drawn from the WGCNA analysis were quite complex, thus independent verification of some predictions would be quite valuable. For example, consider the passage: "In non-vocal learning juvenile females, interestingly LMAN was specialized relative to the AN by the same gene modules as in males (B, F, and I) as well as an additional module G (Fig. 2b); RA was specialized by module A as in males, but not module L and by additional modules A and G. In contrast, neither juvenile female HVC nor Area X exhibited significant gene module expression specializations relative to their surrounds." Providing in situ hybridization verification of these regional gene expression predictions with a few representative genes seems quite feasible given the group's expertise and would considerably strengthen confidence in the module-based inferences.

We performed in-situ independent validation of 36 candidate genes in our first study with this dataset (Choe et al 2021). We now mention this validation in the revised paper. The reviewer’s selection of one of our sentences though made us realize that our grammar used to explain the results was not as clear as it needs to be. We thus cleaned up the grammar of our module descriptions so that it should be communicated with less complexity, the main issue noted by the reviewer.

Because this is a re-analysis of a previously published dataset, the authors should more explicitly describe somewhere in the Discussion how the present analysis advances the understanding of sex differences in songbird neuroanatomy and behavior beyond the previous analysis.

We have added an additional sentence into the discussion more clearly separating the results of the current study from our previous work.

Specific comments:Abstract:There is evidence (from Frank Johnson's lab) that RA does not completely atrophy in female zebra finches, but is still present with more preserved connectivity than previously thought, possibly related to non-singing function(s). A term like 'marked reduction' of female RA may more accurately reflect the current state of knowledge.

We have changed the text to “partial atrophy”.

The term "driver" is undefined and unclear at this point of the paper; a clear definition for "driver" is also lacking in the Intro.

We now define “driver” or “genetic driver” as understood to mean “a genetic locus whose expression and/or inheritance strongly regulates the trait of interest”.

When citing the literature on studies that identified "specific genes with specialized up- or down-regulated expression in song and speech circuits relative to the surrounding motor control circuits", the authors should also cite studies from other labs (e.g. Li et al., PNAS, 2007; Lovell et al, Plos One 2008; Lovell et al, BMC Genomics 2018; Nevue et al, Sci Rep. 2020), to be accurate and fair.

Citations added

For clarity, the authors should explicitly formulate the hypothesis they are proposing at the end of the Summary.

We thank the reviewer for this comment. We have replaced the final sentence of the summary with: “We present a hypothesis where reduced dosage and expression of these Z chromosome genes changes the developmental trajectory of female HVC, partially preventable by estrogen treatment, contributing to the loss of song learning behavior.”

Introduction:Vocal learning is arguably the ability to imitate 'vocal' sounds, this could be clarified here.

We have amended the sentence to “Vocal learning is the ability to imitate heard sounds using a vocal organ…”

Given they are currently considered sister taxa, can the author briefly explain what is the basis for assuming that songbirds and parrots independently evolved vocal learning?

Although songbirds and parrots belong to a monophyletic clade, they are not sister taxa. There are two clades separating them that are vocal non-learners. We have cited the reference that demonstrated this (e.g. Jarvis et al 2014 Science).

Why use Taeniopygia castanotis rather than the more broadly used Taeniopygia guttata?

Zebra finches were recently reclassified and T.castanotis is now more accurate. The Indonesian Timor zebra finch retained T.guttata while the Australian finch, used here, was classified as T.castanotis.

The authors state: "...vocal learning is strongly sexually dimorphic in zebra finches and many other vocal learning species" and cite Nottebohm and Arnold, Science, 1978. That landmark paper only shows dimorphism in song nuclei (not learning) in two songbird species. The authors should provide citations for other species and behavior, or modify the statement.

We have added an additional citation (Odom et al.) to this sentence which covers the phylogeny more broadly.

The authors refer to the nucleus RA as being located in the lateral intermediate arcopallium (LAI). Other labs have described this domain as the dorsal part of the intermediate arcopallium, thus AId or AID (Mello et al., JCN, 2019; Yuan and Bottjer, J Neurophys 2019; Yuan and Bottjer, eNeuro, 2020; Nevue et al., BCM Genomics, 2020). The authors should acknowledge this discrepancy in nomenclature so that data and conclusions can be more readily compared across studies.

We thank the reviewer and agree that this is helpful. We have added a note at the first mention of LAI.

The authors state that data from the gynandromorph bird described by Agate et al implicates "sex chromosome gene expression within the song system" as involved in the song system sexual dimorphism. That study, however, only rules out circulating gonadal steroids, and while suggesting a cell-autonomous mechanism like sex chromosome genes, it does not necessarily exclude other brain-autonomous factors like sex differences in local production of sex steroids.

We say that this study “implicated” sex chromosome gene expression, which is accurate per the results and discussion of that study. We are unsure what “brain autonomous factors like sex differences in local production of sex steroids” means?. “Brain autonomous” and “local production” in the brain seem contradictory in this context?

Results:The authors state that "the E2-treated females in this study had similarly sized song system nuclei as males, indicating that E2 treatment prevented atrophy". Can they clarify whether the VEH-treated females actually had smaller RAs than E2-treated females or VEH-treated males at this age? This is still quite early in development and it is unclear to what extent RA's marked sexual dimorphism in adults or later developmental ages has already taken place in untreated (or VEH-treated) birds. A related comment is that the authors state later on: "We interpret these findings to indicate that: LMAN and RA atrophy later in juvenile female development..." Does this mean these nuclei actually did not show the marked decreases predicted earlier in the text? Clarifying this point would be helpful.

We thank the reviewer for pointing out this discrepancy, which reviewer #1 asked for clarification as well. RA size at this age is similar in males and females. However, HVC and Area X is smaller and absent respectively in females and E2 treatment partially prevents this atrophy. The text now reads:

“In our previous study, we found that estradiol treatment in PHD30 females caused HVC to enlarge and Area X to appear when it normally does not develop in females, but both at sizes less than in untreated or treated males.The sizes of PHD30 female LMAN RA were already the sizes as seen in males, as the later has not atrophied yet at this age(25).”

The authors acknowledge that area X is absent in untreated and VEH-treated females. Could they please clarify how area X and the surrounding stratal tissue that excludes area X were identified for laser capture dissections in juvenile females?

We have added the following statement to the main text portion discussing the dissections.

“In the case of vehicle-treated females which lack Area X, a piece of striatum from the same location of where Area X is found in males was taken. “

Some passages in Results discussing the authors' interpretation of the modules seem quite speculative and possibly belong instead in the Discussion. For example: "... that module A and G genes could be associated with the start of this atrophy; HVC and Area X are likely the first to atrophy or not develop; and lack of any gene module specialization in them at this age could mean that they would be more sensitive to estrogen prevention of vocal learning loss."

As suggested, we have removed this text from the results; these ideas were already presented in the Discussion. We have merged the resulting small paragraph with the preceding paragraph.

The authors state: "To assess the effects of chronic exogenous estrogen on the developing song system, we first performed a control analysis of modules in the E2-treated juvenile males." How can an assessment of estrogen effects be a "control" analysis? Does this refer to a contrast with females? Please clarify the language here.

The reviewer is correct, that E2 treatment in males should not be considered a control experiment. We removed the word “control”.

When discussing the GO-enriched terms for module G, it is unclear how the authors reached the conclusion about "proliferative", as the enriched terms do not refer to processes more directly indicative of proliferation like "cell division" or "cell cycle regulation". Rather, these terms seem more related to differentiation and growth, which do not necessarily imply proliferation. The authors also refer to "HVC proliferation" later on in the Discussion. However, there is conclusive evidence from several labs that proliferative events associated with postnatal neuronal addition and/or replacement in song nuclei occur in the subventricular zone, not in song nuclei like HVC itself, and that the growth of song nuclei largely reflects cell survival, as well as growth in size and complexity under the regulation of sex steroids.

We agree that “proliferative” may have been a poor word choice here. We did not mean to indicate that cell division was occuring in HVC itself. Instead we meant to indicate that HVC is able to accommodate the new born neurons from the SVZ. We have replaced the word “proliferative” throughout. In the instance the reviewer mentions specifically we replaced it with,“...potentially act to integrate and differentiate late born neurons.”

With regard to module E, referring to a telencephalon-wide sexually dimorphic gene expression program seems quite a stretch, given that only a few regions were sampled and compared between sexes. These related statements should be toned down.

We have replaced “telencephalon-wide” with “more distributed across the finch telencephalon” and other similar language in each instance.

The following passage is very speculative and should shortened and/or moved to the Discussion: "Based on the findings in these gene sets, we hypothesize that without excess estrogen in females, HVC expansion is prevented by not specializing the growth and neuronal migration promoting genes in module G to the HVC lineage by late development. This is potentially enacted by depleting necessary gene products from the Z sex chromosome, such as GHR, which are already present in only one copy."

We have deleted this portion of the text, as the idea is already present in the discussion.

Figure 5: To this reviewer, the comparisons of sex differences and of female response to E2 are the most relevant and informative ones, whereas the regional differences between song nuclei and surrounds refer to different cell populations and cell types where other processes may be occurring, independently of what occurs in song nuclei. It thus seems like the intersection analysis in panel 5i may be subtracting out important "core genes" in terms of E2 effects and/or sex differences in the most relevant cell populations, i.e. in this case within song nucleus HVC.

Song learning and the vocal learning brain regions are specialized behaviors and associated nuclei which have a set of hundreds of specialized genes compared to the surrounds. Our previous findings shows that E2 drives the appearance of these specializations in female zebra finches. Thus, we considered this the most interesting question to focus on, which we have further highlighted. Nevertheless, in response to the reviewers suggestion, we have added a .xlsx supplemental file containing the results from each of the individual tests so readers may examine any single comparison, or set of comparisons, in more detail.

Discussion:It is unclear what the term "critical period" refers to in: "during the critical period of atrophy for the female vocal circuit"; please clarify.

We agree that our language was nebulous. We have replaced it with “as several male song control nuclei begin to expand and female nuclei partially atrophy”

In: "HVC appeared unspecialized at the level of gene module expression in control females", does "unspecialized" refer to a lack of difference in gene expression when compared to surroundings? Please clarify. The same comment applies to other uses of "unspecialized" in this paragraph.

Yes, unspecialized means lack of difference in gene expression in the song nucleus. To clarify this point, we have reworked that and the following sentence as follows:

“HVC appeared unspecialized compared to the surrounding nidopallium at the level of gene module expression in control females, with no significantly differentially expressed MEGs . However, in E2-treated females, HVC exhibited a subset of the observed male HVC gene expression specializations. Similarly, the vehicle-treated female striatum located where Area X would be also lacked any specialized gene module expression, but the E2-treated female Area X exhibited a subset of the male Area X specializations, consistent with the known absence of Area X in vehicle-treated females and presence in E2-treated females.”

The authors state: "...we surprisingly found that the most specialized genes were disproportionately from the Z chromosome", when discussing module G in HVC. Why is this so surprising? In a sense, this could be taken as consistent with the findings of Friedrich et al, 2022, where sex differences in the RA transcriptome were predominantly Z related on 20 dph. Arguably 20 dph is still quite close to 30 dph in the present study, when compared to 50 dph in Friedrich et al, when autosomes predominate.

Our bioRxiv was originally posted in July 2021, prior to the publication of Friedrich et al, 2022; however we had previously added to our discussion that several of our results are consistent with the observations of Friedrich et al..

We have a different interpretation of Z chromosome gene results in Friedrich et al.. While the percentage of specialized genes from the Z chromosome decreased, the absolute number of specialized Z chromosome genes actually increased over this interval. In Fig. 3a from Friedrich et al. it appears that ~28% of Z chromosome genes were sexually dimorphic in their expression in RA at PHD20 but that ~39% of Z chromosome genes were similarly dimorphic at PHD50. We interpret this result as the Z chromosome genes being among the earliest genes differentially expressed between the sexes, not that their differential expression or role ever subsequently decreased. We have reworked this portion of the discussion to make our point more clear:

“This model of sex chromosome influenced song system development is consistent with recent observations comparing male and female zebra finch transcriptomes from RA at young juvenile (PHD20) and young adult (PHD50) ages in un-manipulated birds (Friedrich et al. 2022)57. While that study proposes that the role of the sex chromosome in maintaining transcriptomic sex differences diminishes across development, as the proportion of specialized genes that originate on the sex chromosomes diminishes, this effect was driven by large increases in differentially expressed autosomal genes rather than by any reduction in sex chromosome dimorphism; the percentage of differentially expressed Z chromosome genes increased from PHD20 (28%) to PHD50 (39%) (Friedrich et al). This leads us to conclude that sexually dimorphic Z chromosome expression at juvenile ages precedes the sexually dimorphic expression of the autosomes seen in adults. This is consistent with our hypothesis that sufficient expression of select Z chromosome gene products (GHR, etc..) is necessary for subsequent autosomal song system specializations (modG).”

Further, when we write ”When examining the module G HVC specialization induced by E2-treatment in female HVC, we surprisingly found that the most specialized genes were disproportionately from the Z chromosome” we are referring to the upregulation of module G by E2 in female HVC, not the sex difference described in RA by Friedrich et al. which only utilized un-treated RA samples and thus is more likely related to our observations of module E.

The term "sexual dimorphism" has been more traditionally used for sex differences that are very marked, like features that are highly regressed or absent in one sex, most often in females. Quantitative differences in gene expression, including dosage differences like those related to module E, are more appropriately described as sex differences rather than dimorphisms. That usage would be more consistent with most of the literature, and thus preferable.

We did a google search for common definitions, and found more the opposite. Sexual dimorphism being used more often as differences of degree (with the zebra finch example as one of the top hits), and sex differences being used often as more absolute differences (like presence vs absence of the Y chromosome). Further, as in the reviewer’s first sentence, the definition of sexual dimorphism is a sex difference. That is, the two phrases can be interchangeable. Thus, we prefer to keep sexual dimorphism.

Several references are incomplete or seem truncated, like 9 and 10.

Fixed

Table S2: Please examine and take into account the W gene curation presented in Table S3 of Friedrich et al., 2022.

We have added additional supplementals (supplemetal_w_chrom_express.csv and supplemetal_z_chrom_express.csv) of the data provided in new Fig 5 incorporating the curation information from Table S3 from Friedrich et al.

Data availability:Genes for all the main modules identified should be presented in a Supplemental Table, or through a link to a stable data repository.

We have added an additional Supplemental Table supplemental_gene_module_assignment.csv with this information.